# RSPO: Regularized Self-Play Alignment of Large Language Models

## Abstract

Self-play alignment has emerged as an effective approach for fine-tuning large language models (LLMs), formulating preference optimization as a two-player game. However, the regularization with respect to the reference policy, which is crucial for mitigating over-optimization, has been insufficiently investigated in self-play alignment. To study the impact of different regularization strategies, we propose **Regularized Self-Play Policy Optimization (RSPO)**, a novel framework that unifies prior methods and enables simple plug-and-play regularizers, meanwhile preserving convergence to Nash equilibrium of the corresponding regularized game. We observe that RSPO with appropriate regularizers can substantially improve the length-controlled win rate (LCWR) on AlpacaEval-2 across a range of base models, while also achieving consistently superior performance on Arena-Hard, MT-Bench, ArmoRM, and response diversity. In particular, RSPO improves unregularized self-play baseline (SPPO) on AlpacaEval-2 LCWR from $28.5\%$ to $35.4\%$ with base model Mistral-7B, from $38.77\%$ to $43.66\%$ with LLaMA-8B, and from $50.54\%$ to $51.83\%$ with Gemma-2B. Combining simplicity, convergence guarantees, and significant empirical gains, RSPO offers a strong foundation for exploring regularized self-play in language model alignment.

## 1 Introduction

Self-play is a line of work conducting iterative self-competition of models, which has been demonstrated as an effective approach for improving AI systems (Goodfellow et al., 2020; Wang et al., 2022), particularly in strategic decision-making problems (Silver et al., 2016; Heinrich & Silver, 2016; Pinto et al., 2017; Brown & Sandholm, 2018). In the human alignment of LLMs, self-play recently started to be used and has shown superior empirical performance than other iterative Reinforcement Learning from Human Feedback (RLHF) methods on popular benchmarks (Dubois et al., 2024; Jiang et al., 2024; Wu et al., 2024; Rosset et al., 2024). By formulating the preference optimization problem as a two-player game, self-play alignment methods seek to identify a *Nash Equilibrium* (NE) of the game in which utility is determined by a general preference model (Azar et al., 2024; Munos et al., 2023; Calandriello et al., 2024). This NE is regarded as the most aligned LLM policy achieved without Bradley-Terry (BT) reward modeling (David, 1963), which has shown under-performance compared to general preference modeling (Ye et al., 2024).

Despite the significant empirical improvements achieved through self-play, the impact of regularization to the reference policy—commonly used in RLHF to mitigate over-optimization—has received insufficient investigation in self-play alignment. Most existing self-play methods completely lack explicit regularization (Wu et al., 2024; Rosset et al., 2024; Swamy et al., 2024; Wang et al., 2024b; Gao et al., 2024). In practice, unregularized self-play is also susceptible to over-optimization, particularly when the preference model is inaccurate or misspecified. Although a few recent self-play approaches like Nash-MD (Munos et al., 2023) incorporate reverse KL divergence as a regularization penalty (Calandriello et al., 2024; Wang et al., 2024b; Zhang et al., 2024b), it remains unclear whether reverse KL is optimal for alignment, and the broader impact of alternative regularization strategies in self-play remains insufficiently explored. Moreover, the extension of current approaches to general forms of regularization is challenging, as their training protocols are intrinsically reliant on the reverse KL divergence for regularization Munos et al. (2023) (see Figure 1).

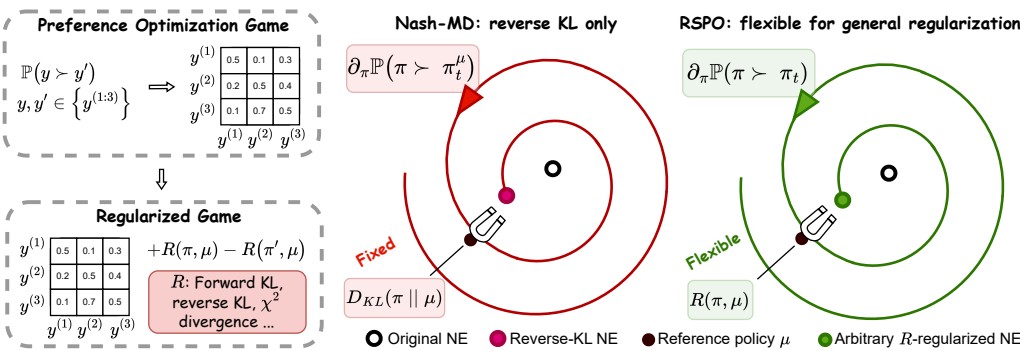

Figure 1: **RSPO is flexible for general regularization.** The estimation of Nash-MD policy update direction $\partial_\pi \mathbb{P}(\pi \succ \pi_t^\mu)$ requires samples from geometric mixture policy $\pi_t^\mu$. Such update approach is only compatible with reverse KL divergence for regularization.

In this work, we introduce a novel framework to flexibly incorporate diverse regularization methods into self-play alignment, termed **Regularized Self-Play Policy Optimization (RSPO)**:

- RSPO offers a simple way to apply general regularization strategies in self-play by **directly adding** the regularization term to our proposed unified self-play loss function, while maintaining **last-iterate convergence** to NE of the corresponding regularized preference optimization game. Unlike Nash-MD, which requires a specialized sampling process limited to Reverse-KL regularization, our method follows the standard sampling procedure in RLHF, making it both simpler and more general.

- RSPO with tuned regularizers demonstrates substantial improvements over the unregularized self-play alignment method (SPPO (Wu et al., 2024)). In particular, it increases the length-controlled win rate on AlpacaEval-2.0 **from** $28.5\%$ **to** $35.4\%$ **with Mistral-7B, from** $38.77\%$ **to** $43.66\%$ **with LLaMA-8B, and from** $50.54\%$ **to** $51.83\%$ **with Gemma-2B**. RSPO also achieves consistently superior performance on other benchmarks, including Arena-Hard-v0.1, MT-Bench, self-BLEU diversity (Zhu et al., 2018), and ArmoRM across multiple reward dimensions such as instruction following, truthfulness, honesty, and helpfulness.

- Empirical analysis reveals distinct effects of different regularizations. On both Mistral-7B-Instrct and LLaMA-8B-Instruct stronger forward KL regularization reduces the response length, whereas reverse KL regularization significantly improves the raw win rate. Mistral-7B-Instrct with combined forward and reverse KL regularization achieves the most improvement. In addition, RSPO also demonstrate parameter-efficiency when comparing with SPPO trained with stronger preference model, indicating the comprehensive effectiveness of our method in self-play alignment.

## 2 PRELIMINARIES

We denote a prompt as $x$, a response as $y$, and a LLM policy as $\pi(y|x)$, where $\pi(\cdot|x) \in \Delta_\mathcal{Y}$. We denote the set of all prompts as $\mathcal{X}$, and the set of all responses as $\mathcal{Y} = \{y^0, y^1, \cdots\}$. We use $\Delta_\mathcal{Y}$ to denote the probability simplex over the responses given a specific prompt. We parametrize the LLM policy $\pi$ as $\pi_\theta$. The reference policy is an LLM denoted as $\mu \in \Delta_\mathcal{Y}^\mathcal{X}$. For notational brevity, we remove the dependence of policy $\pi$ and loss functions on the prompt $x$ throughout the paper.

### 2.1 GAME-THEORETIC PREFERENCE OPTIMIZATION

We study the preference optimization problem in an online setting by formulating it as a two-player max-min game, as studied in previous self-play works (Wu et al., 2024). The players are two LLMs whose strategies are LLM policies, denoted as max-player $\pi$ and min-player $\pi'$. The utility of the max-player is expressed as the preference of itself over the min-player:

$$u(\pi; \pi') = \mathbb{P}(\pi \succ \pi') \stackrel{\text{def}}{=} \mathbb{E}_{y\sim\pi, y'\sim\pi'}[\mathbb{P}(y \succ y')], \tag{1}$$

where $u : \Delta_\mathcal{Y}^\mathcal{X} \times \Delta_\mathcal{Y}^\mathcal{X} \to \mathbb{R}$ is *linear* in $\pi$ and $\pi'$; $\mathbb{P} : \mathcal{X} \times \mathcal{Y} \times \mathcal{Y} \to [0, 1]$ is a general preference model that quantifies the preference of $y$ over $y'$ given a prompt. We extend the notation $\mathbb{P}(y \succ$

$\pi') = \mathbb{E}_{y' \sim \pi'}[\mathbb{P}(y \succ y')]$. The objective is finding a *NE* policy $\pi^*$ of the preference model:

$$(\pi^*, \pi^*) = \arg\max_{\pi} \min_{\pi'} \mathbb{P}(\pi \succ \pi'). \tag{2}$$

Therefore, an NE strategy $\pi^*$ is an LLM that can generate *the most preferred responses in expectation*, thus achieving human alignment based on the preference model. Most existing self-play alignment methods aim to solve this NE following Algorithm 1 (Wu et al., 2024; Rosset et al., 2024; Swamy et al., 2024; Wang et al., 2024b).

## 2.2 PREFERENCE OPTIMIZATION VIA MULTIPLICATIVE WEIGHTS UPDATE

An effective self-play method to solve the preference optimization game in Equation (2) is Self-Play Policy Optimization (SPPO) (Wu et al., 2024). SPPO derives its loss function from the iterative no-regret learning algorithm, Multiplicative Weights Update (MWU) (Freund & Schapire, 1997). Specifically in a game setting, denote learning rate as $\eta$, and normalization constant $Z(\pi_t)$. In any iteration $t$, the policy update $\forall y \in \mathcal{Y}$ is $\pi_{t+1}(y) = \pi_t(y) \cdot \exp\left(\eta \mathbb{E}_{y' \sim \pi_t}[u(y; y')]\right)/Z(\pi_t)$, where $u(y; y')$ is the utility function defined in Equation equation 1, with $y$ treated as a pure strategy.

The practical loss function of SPPO for policy update is then derived according to MWU:

$$\mathcal{L}_{\text{SPPO}}(\theta) = \mathbb{E}_{y \sim \pi_t}\left[\log \frac{\pi_\theta(y)}{\pi_t(y)} - \left(\eta \mathbb{P}(y \succ \pi_t) - \log Z(\pi_t)\right)\right]^2. \tag{3}$$

SPPO converges to the NE of the preference optimization game in Equation (2). However, after running multiple iterations, the deviation of the policy $\pi_\theta$ from $\mu$ can be large. Such deviation is particularly problematic when the preference model is only accurate at evaluating responses sampled from the reference policy (Munos et al., 2023). Furthermore, in aligning LLMs in practice, the preference model is typically a surrogate $\hat{\mathbb{P}}$, such as PairRM (Jiang et al., 2023b), which may be misspecified at some out-of-distribution responses and inaccurate due to estimation error or limited model expressiveness (PairRM is only a 0.4B model), causing over-optimization problem. Regularizing the policy optimization to a reference SFT model, which is typically trained on high-quality data (Ouyang et al., 2022), can mitigate the problem. We provide a synthetic example in Appendix D.1 to demonstrate this problem.

## 2.3 REGULARIZED PREFERENCE OPTIMIZATION GAME WITH REFERENCE POLICY

To address the regularization in self-play, we adopt the objective in Nash Learning from Human Feedback (Munos et al., 2023), and extend the KL divergence regularization to a general regularization function, to penalize the deviation from the reference policy. We define a *convex* regularization function $R : \Delta_{\mathcal{Y}}^{\mathcal{X}} \times \Delta_{\mathcal{Y}}^{\mathcal{X}} \to (-\infty, \infty)$, where $R(\pi, \mu)$ measures the distance between $\pi$ and the reference model $\mu$, such as KL divergence $D_{\text{KL}}(\pi\|\mu)$. Denote regularization temperature as $\tau$, the objective becomes to optimize a *regularized preference model* by solving the NE $(\pi^*, \pi^*)$ of the *regularized* game, where the utility of max player is still $u(\pi; \pi') = \mathbb{P}(\pi \succ \pi')$:

$$\arg\max_{\pi} \min_{\pi'} \mathbb{P}(\pi \succ \pi') - \tau R(\pi, \mu) + \tau R(\pi', \mu). \tag{4}$$

We provide proof of the existence and uniqueness of this NE in Appendix A.1. A few recent methods leverage Mirror Descent (MD), which is also in a self-play manner, to find a regularized NE in Equation (4) with last-iterate policy (Munos et al., 2023; Calandriello et al., 2024; Zhang et al., 2024b).

However, these MD-based methods are only compatible with the reverse KL divergence regularizer, and are non-trivial to extend to general divergence. For instance, Nash-MD[1] addresses the reverse KL regularization of $\pi$ and $\mu$ requiring responses generated from a geometric mixture policy $\pi_t^\mu(y) \propto \pi_t(y)^{1-\eta\tau}\mu(y)^{\eta\tau}$ (Munos et al., 2023), which is inherently compatible only with reverse KL divergence:

$$\pi_{t+1} = \arg\min_{\pi} -\eta \mathbb{E}_\pi[\nabla_\pi u(\pi_t; \pi_t^\mu)] + D_{\text{KL}}(\pi\|\pi_t^\mu). \tag{5}$$

Therefore, while the LLMs optimized via existing self-play methods exhibit empirical improvement, they all have limited regularization of $\pi$ and $\mu$. The potential benefits of alternative regularization, such as adopting other $f$-divergences than reverse KL, remain unexplored.

---

[1]Throughout the paper, regularization specifically refers to the deviation of $\pi$ from $\mu$, rather than from $\pi_t$.

## 3 REGULARIZED SELF-PLAY POLICY OPTIMIZATION

We propose a framework for regularized self-play alignment, namely **Regularized Self-Play Policy Optimization (RSPO)**. RSPO is simple and flexible for regularization, and provably convergent to Nash Equilibrium. The loss function of RSPO $\mathcal{L}_{\text{RSPO}}$ is defined as the sum of a mean-squared self-play loss and a weighted regularization term:

$$\mathcal{L}_{\text{RSPO}}(\theta; G, B, R) \stackrel{\text{def}}{=} \mathbb{E}_{y \sim \pi_t} \Big[ \log \frac{\pi_\theta(y)}{\pi_t(y)} - \eta \Big( G(y, \pi_t, \mu) - B(\pi_t, \mu) \Big) \Big]^2 + \lambda R(\pi_\theta, \mu), \quad (6)$$

where $G(y, \pi_t, \mu)$, $B(\pi_t, \mu)$, and $R(\pi_\theta, \mu)$ are configurable components. First, $G : \mathcal{Y} \times \Delta_{\mathcal{Y}}^{\mathcal{X}} \times \Delta_{\mathcal{Y}}^{\mathcal{X}} \to (-\infty, \infty)$ defines the *update direction* of $\pi_\theta$, which can be set as the gradient of a utility function to guide the policy update towards increasing the utility. Second, the *baseline* function $B : \Delta_{\mathcal{Y}}^{\mathcal{X}} \times \Delta_{\mathcal{Y}}^{\mathcal{X}} \to (-\infty, \infty)$ is for variance-reduction of $G$, similar to the baseline in REINFORCE (Williams, 1992). Lastly, $R : \Delta_{\mathcal{Y}}^{\mathcal{X}} \times \Delta_{\mathcal{Y}}^{\mathcal{X}} \to \mathbb{R}$ is the regularization function. The coefficient $\lambda$ is the regularization temperature. The first Mean Square Error term in Equation (6) can be interpreted as a self-play loss of conducting exponentiated gradient descent (Beck & Teboulle, 2003).

RSPO is a modular framework offering a simple way to introduce regularization into self-play alignment with *only an additional term in the loss*. RSPO offers the simplicity and flexibility to incorporate *various* regularization methods into self-play-based preference optimization methods. Additionally, we show in Section 3.1 that RSPO can generalize existing unregularized self-play methods without external regularization $R$. Thus, regularizing existing methods requires *no change* to their original loss functions or hyperparameters, but simply adding an external plug-and-play regularization to their loss function and tuning the temperature $\lambda$.

In practice, we set baseline function $B = \frac{1}{2}$ following Nash-MD and SPPO, and the update direction $G$ to be the gradient of the preference against $\pi_t$, $\forall y \in \mathcal{Y}$:

$$G(y, \pi_t, \mu) = \partial_{\pi(y)} \mathbb{P}(\pi \succ \pi_t) = \mathbb{P}(y \succ \pi_t). \quad (7)$$

We execute Algorithm 1 by applying the following RSPO loss with any regularization $R$ of interests:

$$\mathcal{L}_{\text{RSPO}}\big(\theta; G = \mathbb{P}(y \succ \pi_t), B = \tfrac{1}{2}, R\big). \quad (8)$$

In theory, $B$ helps minimize the variance of $G$ the most when $B = \mathbb{E}_{y \sim \pi_t}[G(y, \pi_t, \mu)]$. But in preference optimization, due to the typically small minibatch size, the estimation error of the mean of $G$ could be large, leading to additional estimation error of the loss. Thus, we also set the baseline value for variance reduction to be a constant $\frac{1}{2}$, the mean value of $G$ when the algorithm converged. For the implementation of various divergence-based regularization, refer to Appendix C.3.

In the following sections, we first illustrate the generalizable formulation of RSPO, so that it can be implemented without modifying the existing self-play component. We then establish theoretical convergence guarantees for RSPO grounded in Mirror Descent theory.

### 3.1 GENERALIZING EXISTING SELF-PLAY METHODS

In this section, we show how RSPO generalize existing self-play methods, which showcase (1) implementing RSPO requires only one additional term to existing self-play loss functions; (2) the limitation of existing regularized methods. First, the unregularized self-play method SPPO (Wu et al., 2024) has a loss function defined in Equation (3) equivalent to RSPO *without external regularization*:

$$\mathcal{L}_{\text{SPPO}}(\theta) = \mathcal{L}_{\text{RSPO}}\Big(\theta; G = \mathbb{P}(y \succ \pi_t), B = \frac{1}{2}, R = 0\Big). \quad (9)$$

According to Equation (8) and Equation (9), $\mathcal{L}_{\text{RSPO}} = \mathcal{L}_{\text{SPPO}} + \lambda R(\pi_\theta, \mu)$, i.e. the implementation of RSPO is equivalent to directly add the regularization $R$ to the loss function of SPPO (Equation (3)). This implies that the additional regularization term becomes plug-and-play, requiring minimal changes to existing training pipeline.

In addition, existing regularized methods can be generalized by $\mathcal{L}_{\text{RSPO}}$ (derivations in Appendix A.3):

$$\nabla_\theta \mathcal{L}_{\text{Nash-MD}}(\theta) = \nabla_\theta \mathcal{L}_{\text{RSPO}}\Big(\theta; G = \mathbb{P}(y \succ \pi_t^\mu), B = \frac{1}{2}, R = D_{\text{KL}}(\pi_\theta \| \mu)\Big) \quad (10)$$

$$= \nabla_\theta \mathcal{L}_{\text{RSPO}}\Big(\theta; G = \mathbb{P}(y \succ \pi_t^\mu) - \tau \log \frac{\pi_t(y)}{\mu(y)}, B = \frac{1}{2}, R = 0\Big). \quad (11)$$

Equation (10) verifies our summarization shown in Figure 1. The convergence guarantee of Nash-MD (Munos et al., 2023, Lemma 2) requires the policy updated with Equation (5), which is specifically designed for reverse KL regularization, as other $R$ can not be merged with $D_{\text{KL}}(\pi\|\mu)$ to a regularization w.r.t. geometric mixture $\pi_t^\mu$. Additionally, Equation (11) demonstrates that RSPO enables to even add extra regularization to existing regularized self-play methods.

## 3.2 THEORETICAL GUARANTEES

In this section, we examine the theoretical properties of RSPO, with a particular emphasis on its convergence guarantee. We adopt Mirror Descent (MD) as the foundational framework, given its well-established last-iterate convergence to the NE.

We build upon Magnetic Mirror Descent (MMD) (Sokota et al., 2022), a specialized variant of MD that guarantees convergence to a reverse-KL-regularized NE. To generalize beyond reverse-KL regularization, we introduce Generalized Magnetic Mirror Descent (GMMD), which can accommodate a broader class of regularization techniques. By demonstrating that optimizing the RSPO loss is equivalent to performing reinforcement learning (RL) within the GMMD framework, we establish a formal connection between RSPO and GMMD. This connection ensures the last-iterate convergence of RSPO to the NE of the corresponding *regularized* game.

**Tabular GMMD.** Denote the utility function of the game as $U$, define $G$ as the element of the vector of partial derivatives of $U$ w.r.t. policy:

$$G(y;\pi') \stackrel{\text{def}}{=} \partial_{\pi(y)}U(\pi;\pi'),\ \partial_\pi U(\pi;\pi') = (G(y^0;\pi'),\cdots,G(y^{|\mathcal{Y}|};\pi'))^\top \in \mathbb{R}^{|\mathcal{Y}|} \quad (12)$$

Then in iteration $t$, GMMD updates policy as

$$\pi_{t+1} = \arg\min_\pi -\eta\mathbb{E}_\pi[G(y;\pi_t)] + B_\psi(\pi;\pi_t) + \tau R(\pi,\mu), \quad (13)$$

where $\tau$ is regularization temperature, $R$ is a general regularization function, serving as a "magnet" to attract $\pi$ to $\mu$ during policy updating. $B_\psi$ is the Bregman Divergence generated by a convex potential function $\psi$ (Bregman, 1967).

Notably, the vanilla Magnetic Mirror Descent limits $R$ to be the same regularization method of $\pi$ and $\pi_t$, i.e., $R = B_\psi$ (Sokota et al., 2022, Section 3.2); whereas in this paper we aim at a general regularizer of $\pi$ and $\mu$, which could be different from $B_\psi$, and study the effects of different regularization methods.

**Proposition 3.1** (**Last-iterate Convergence**). *If $R(\cdot,\mu)$ is 1-strongly convex relative to $\psi$, $\eta \leq \tau$, and $U$ is linear, then policy updated by GMMD in Equation* (13) *has last-iterate convergence to the following regularized NE* $\max_\pi \min_{\pi'} U(\pi;\pi') - \tau R(\pi,\mu) + \tau R(\pi',\mu)$.

Proposition 3.1 is a direct application of Theorem 3.4 by Sokota et al. (2022), which guarantees the last-iterate convergence of GMMD to the NE of a regularized game (Proof in Appendix A.4).

**Deep RL Implementation of GMMD.** To adapt GMMD to preference optimization problems, RL techniques are commonly employed as practical implementations, as for many MD update (Tomar et al., 2020; Munos et al., 2023; Wang et al., 2024b). Define the loss function of conducting GMMD in preference optimization as

$$\mathcal{L}_{\text{GMMD}}(\theta) \stackrel{\text{def}}{=} -\eta\mathbb{E}_{\pi_\theta}\big[G(y;\pi_t)\big] + D_{\text{KL}}(\pi_\theta\|\pi_t) + \tau R(\pi_\theta,\mu). \quad (14)$$

Here, we set the Bregman divergence to Reverse KL in preference optimization as in previous works (Munos et al., 2023; Zhang et al., 2024b). The gradient estimation of $\mathcal{L}_{\text{GMMD}}(\theta)$ for policy updates is required since the expectation in the first term is dependent on $\pi_\theta$. Following Policy Gradient (PG) theorem (Sutton et al., 1999), the PG of GMMD is equal to $\nabla_\theta\mathcal{L}_{\text{RSPO}}(\theta)$ up to multiplying a constant:

$$\nabla_\theta\mathcal{L}_{\text{GMMD}}(\theta) = \mathbb{E}_{y\sim\pi_\theta}\left[\nabla_\theta\log\pi_\theta(y)\left(-\eta G(y;\pi_t) + \log\frac{\pi_\theta(y)}{\pi_t(y)} + B\right)\right] + \tau\nabla_\theta R(\pi_\theta,\mu), \quad (15)$$

where $B$ is a baseline function to reduce the variance as in REINFORCE (Williams, 1992). We set $B$ independent to $\theta$ so that adding $B$ does not affect the value of Equation (14), due to $\mathbb{E}_{y\sim\pi_\theta}[\nabla_\theta\log\pi_\theta(y)\cdot\eta B] = \eta B\nabla_\theta\mathbb{E}_{y\sim\pi_\theta}[1] = 0$.

Due to the equivalence between RSPO and GMMD, we provide the convergence guarantee for our practical implementation of RSPO (Equation (13)), to the Nash equilibrium of the regularized preference optimization game as follows (Proof in Appendix A.5).

**Corollary 3.2.** *Self-play following Algorithm 1 with the RSPO loss function in Equation* (8) *and regularizer $R$ satisfying the assumption in Proposition 3.1, has last-iterate convergence to the NE of the regularized preference optimization game, as described in Equation* (4).

RSPO guarantees NE convergence while allowing flexible regularization strategies, making it a robust extension of self-play optimization. In summary, the proposed RSPO framework provides a generalized approach that simplifies the incorporation of regularization into existing self-play methods while maintaining theoretical soundness.

## 4 EXPERIMENTS

In this section, we answer the following important questions of regularization in the self-play alignment of Large Language Models (LLMs) by testing on various popular benchmarks:

- **Q1:** Does regularization improve the performance of self-play alignment (Sec. 4.1)?
- **Q2:** Which regularization method is the most effective in self-play alignment (Sec. 4.2)?
- **Q3:** What additional advantages can be obtained by regularization in self-play (Sec. 4.3)?

**Experiment Setup.** We evaluate our methods mainly on benchmarks AlpacaEval (Dubois et al., 2024), Arena-Hard (Li et al., 2024), and MT-Bench (Zheng et al., 2023), and test the response generation diversity and quality via self-BLEU (Zhu et al., 2018) and ArmoRM (Wang et al., 2024a), respectively. We follow the experiment setup of SPPO and Snorkel-Mistral-PairRM-DPO (Snorkel) (Tran et al., 2023) to examine our regularization methods, where Snorkel is based on iterative DPO and has achieved strong performance on AlpacaEval. We conduct experiments on base model Mistral-7B-Instruct-v0.2, LLaMA3-8B-Instruct, and Gemma2-2B-IT. Since iterative self-play methods require no response data for training, we only use the *prompts of the Ultrafeedback dataset* (Cui et al., 2023), whose size is $\sim 60\text{K}$. Following SPPO and Snorkel, we split the prompts into three subsets and use only one subset per iteration to prevent over-fitting. To understand the later-iterate performance of self-play, in section 4.1, we also train on the single fold of the prompts iteratively. We use a 0.4B response-pair-wise *preference model* PairRM (Jiang et al., 2023b), evaluated as comparable to $10\times$ larger reward/preference models (Cui et al., 2023).

**Implementations and Baselines.** The implementation of self-play methods follows Algorithm 1. In each iteration, given response-pair-wise preference from PairRM and $K = 5$ number of response samples from the current policy, we estimate the policies' preference $\mathbb{P}(\pi \succ \pi_t)$ and regularization via Monte-Carlo estimation to compute the loss function. We replicate SPPO with the default hyper-parameters and extend it to 9 iterations. We implement RSPO as described in Corollary 3.2. The implementation of regularizations in RSPO is demonstrated in Appendix C.3 using the $K$ samples. We report some of the baseline results from the previous papers, including SPPO, Snorkel (Mistral-PairRM-DPO) (Tran et al., 2023), Mistral-7B (Instruct-v0.2) (Jiang et al., 2023a), iterative DPO by Wu et al. (2024), and SimPO Meng et al. (2024). Since the SPPO paper only provides results across 3 iterations (Wu et al., 2024), we replicate SPPO as an important baseline to study the performance across more than 3 iterations.

### 4.1 EFFECTIVENESS OF REGULARIZATION

In this section, we assess the effectiveness of regularization by comparing the performance of unregularized and regularized self-play methods. We first examine the over-optimization issue inherent in practical self-play alignment by extending the execution of SPPO to Iteration 5. As depicted in Figure 2 (left), a decline in performance appears during the later iterations of SPPO. We hypothesize that this behavior arises from the practical challenges associated with over-optimization.

We then present comprehensive results across three widely used benchmarks (Table 1). RSPO with forward and reverse KL regularization, consistently outperforms the unregularized baseline (SPPO)[2], and iterative DPO in iteration 3, and the strong offline method SimPO across all benchmarks, with a clear performance margin. These results underscore the importance of incorporating regularization

---

[2]We report our replicated testing of the published SPPO Iter3 model (link) on Arena-Hard benchmark. Thus, it is different from the result presented in the original paper of SPPO (Wu et al., 2024).

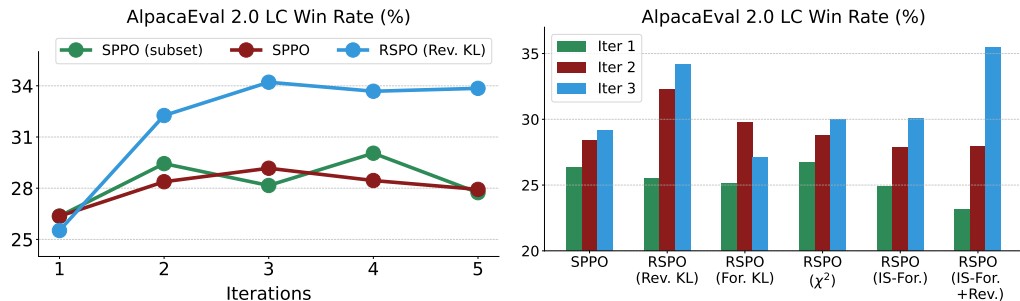

Figure 2: **Left:** Length-controlled win rate (LCWR) across iterations of: SPPO, SPPO trained on a subset of the data: SPPO (subset), and reverse-KL regularized RSPO: RSPO (Rev. KL). SPPO starts to degrade from iteration 3 due to overoptimization, RSPO with reverse KL regularization mitigates it. **Right:** SPPO compared to RSPO with various regularization methods. IS-For.+Rev. regularization in RSPO perform the best. The results are obtained by training on base model Mistral-7B-Instruct.

Table 1: **Comparisons on three popular benchmarks** of baselines, and our strongest model. RSPO with Importance-Sampling-based Forward KL ($\lambda_1 = 0.1$) + Reverse KL ($\lambda_2 = 0.5$) divergence as regularization outperforms baselines on all benchmarks with a clear margin.

| Methods
(Base Model: Mistral-7B-Instruct) | AlpacaEval-2
LCWR ($\%$) | Arena-Hard
Auto-v0.1 | MT-Bench |
|---|---|---|---|
| Mistral-7B-Instruct (Jiang et al., 2023a) | 17.1 | 12.6 | 7.51 |
| Snorkel (Iterative DPO) (Tran et al., 2023) | 26.4 | 20.7 | 7.58 |
| SPPO Iter3 (Wu et al., 2024) | 28.5 | 19.2 | 7.59 |
| SimPO (Meng et al., 2024) | 32.1 | 21.0 | 7.60 |
| RSPO (IS-For.+Rev.) Iter3 | **35.4** | **22.9** | **7.75** |

into self-play alignment. We hypothesize that the effectiveness of regularization arises from the continued utility of the reference policy during optimization, which provides stable guidance and helps mitigate inaccuracies or misspecifications in the general preference model (PairRM).

In Table 2, we further contrast the performance dynamics across iterations of SPPO, other iterative methods, and RSPO (For.+Rev.), regularized by the linear combination of Forward KL and Reverse KL divergence with temperatures of $0.1$ and $0.5$, respectively. The comparative results reveal that regularization significantly improve iterative alignment methods. To rule out the possibility of insufficient iterations affecting performance, we report the best result among nine iterations of our replicated SPPO in Table 2, denoted as "SPPO$^{(3)} \leq 9$", where $(3)$ represents that the strongest model is SPPO-Iter3. SPPO$^{(3)} \leq 9$ consistently underperforms the RSPO. This observation emphasizes that extended training under the unregularized framework fails to match the performance gains achieved through regularization, thereby affirming again the critical role of regularization, and the policy update guidance provided by reference $\mu$ in self-play methodologies.

| Model | AlpacaEval 2.0 | | |
|---|---|---|---|
| | LC Win Rate | Win Rate | Avg. Len |
| Mistral-7B | 17.11 | 14.72 | 1676 |
| Snorkel | 26.39 | 30.22 | 2736 |
| SimPO | 32.1 | 34.8 | 2193 |
| DPO Iter1 | 23.81 | 20.44 | 1723 |
| DPO Iter2 | 24.23 | 24.46 | 2028 |
| DPO Iter3 | 22.30 | 23.39 | 2189 |
| SPPO Iter1 | 24.79 | 23.51 | 1855 |
| SPPO Iter2 | 26.89 | 27.62 | 2019 |
| SPPO Iter3 | 28.53 | 31.02 | 2163 |
| SPPO$^{(3)} \leq 9$ | 29.17 | 29.75 | 2051 |
| RSPO Iter1 | 23.16 | 21.06 | 1763 |
| RSPO Iter2 | 27.91 | 27.38 | 1992 |
| RSPO Iter3 | **35.44** | **38.31** | 2286 |

Table 2: **AlpacaEval LCWR of iterative methods.** RSPO with IS-For.+Rev. regularization shows fast improvement over iterations.

We conduct a comprehensive evaluation of regularization effectiveness across multiple base models and examine scalability to larger preference models, as illustrated in Figure 3. Following the

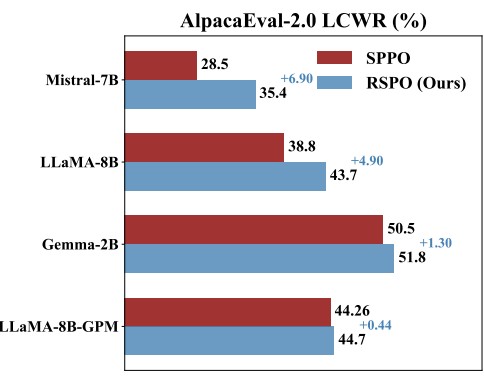
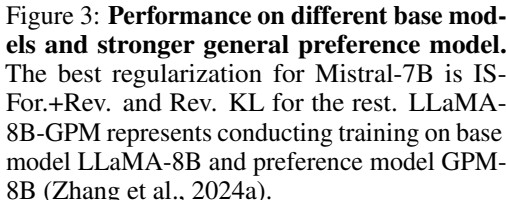

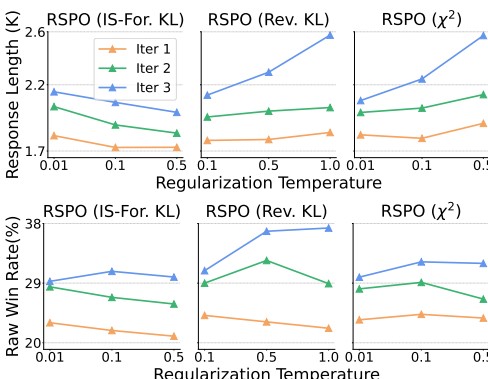

Figure 3: **Performance on different base models and stronger general preference model.** The best regularization for Mistral-7B is IS-For.+Rev. and Rev. KL for the rest. LLaMA-8B-GPM represents conducting training on base model LLaMA-8B and preference model GPM-8B (Zhang et al., 2024a).

Figure 4: **Impact of different regularizations**: also an ablation study on regularization temperature $\lambda$ of RSPO conducted on Mistral-7B. We evaluate how the average response length and raw WR are affected by the stronger regularization. Higher temperature of forward KL leads to shorter response length.

methodology established with Mistral-7B, we identify optimal regularization and hyperparameter for additional base architectures. Our experimental results demonstrate that RSPO consistently outperforms SPPO across all evaluated base models, establishing the robustness of our approach. We also evaluate performance of RSPO trained on a stronger preference model (PM) called GPM-8B (Zhang et al., 2024a). While the performance improvement with the larger preference model is modest (0.44), RSPO demonstrates a significant advantage in parameter efficiency. Notably, RSPO achieves comparable performance to SPPO while utilizing substantially fewer parameters. Specifically, RSPO-LLaMA-8B paired with pairRM-0.5B requires only 8.5B total parameters, achieving performance similar to SPPO-LLaMA-GPM, which utilizes 16B total parameters. This represents a 47% reduction in model size while maintaining competitive performance, highlighting the parameter efficiency.

## 4.2 IMPACT OF DIFFERENT REGULARIZATIONS

We then study the effect of applying different regularization $R$ in RSPO. To obtain a well-regularized self-play, the tuning of regularization temperature $\lambda$ is necessary. An ablation study of the regularization temperature of different methods is shown in Figure 4. According to the figure, the response length increases along with the temperature in reverse KL divergence and Chi-square divergence regularized RSPO. However, both the length and win rate are decreased with stronger regularization via Forward KL divergence, implemented using importance sampling. We attribute the decreasing win rate to the violation of relative convexity assumption (A.1), and the length reduction to the intrinsic mass-averaging property of forward-KL divergence divergence when used for regularization.

In particular, the raw win rate analysis highlights reverse KL divergence as a crucial factor in enhancing self-play performance. We attribute the observed effect to the inherent mode-seeking behavior of reverse KL divergence. Given that forward KL divergence tends to reduce response length while reverse KL divergence yields significant improvements, we adopt a linear combination of both. This approach is designed to balance their complementary effects, ultimately optimizing for a higher LCWR (RSPO (IS-For. + Rev.) in Figure 2 (Right). The hyperparameters are provided in Table 5. Training on other base model also shows similar pattern: forward KL reduce length and reverse KL improve performance (See Table 10 and 11)

## 4.3 RESPONSE DIVERSITY AND OTHER ASPECTS

We demonstrate additional advantages introduced by regularization on Mistral-7B (see also Table 12). Here we demonstrate the diversity of the response cause by regularization. We first provide a motivating example with synthetic data in Appendix D.2, which shows that the unregularized self-play may converge to a collapsed response when multiple equally good responses exist. On the

contrary, RSPO with maximum entropy regularization has multi-modal distribution for generation. For LLMs, we investigate the diversity of generations by estimating the variability of the responses. We use the Self-BLEU (Zhu et al., 2018) score, where a lower score implies higher response diversity. We take the first 200 tokens of each of the 16 generated responses using the prompts of AlpacaEval.

The trend of Self-BLEU scores presented in Table 3 (Right) show that applying RSPO with Reverse KL increases response diversity the most, as well as the LCWRs of AlpacaEval 2.0. Although reverse KL regularization is typically associated with reduced diversity, it can, counterintuitively, enhance diversity when the high-probability region of the reference policy $\mu$ contains multiple modes—a scenario commonly arising when $\mu$ is pretrained on a diverse dataset. In such cases, the sampling-regularized optimization process with reverse KL can also induce additional modes in the learned policy distribution, thereby promoting greater diversity in responses. In contrast, IS-Forward KL yields slightly lower diversity, as its importance sampling–based implementation necessitates hard clipping for numerical stability. Compared to reverse KL, the $\chi^2$ divergence functions as a stronger regularizer (Huang et al., 2024), promoting diversity, albeit at a slower rate.

| Regularization | Iteration | AlpacaEval 2.0 Dataset | |
| --- | --- | --- | --- |
| | | LCWR $\uparrow$ | Self-BLEU $\downarrow$ |
| $\times$ | 1 | 24.79 | 0.751 |
| | 2 | 26.89 | 0.754 |
| | 3 | 28.53 | 0.758 |
| IS-Forward KL + Reverse KL | 1 | 23.16 | 0.747 |
| | 2 | 27.91 | 0.743 |
| | 3 | **35.44** | 0.714 |
| Reverse KL | 1 | 25.52 | 0.747 |
| | 2 | 32.26 | 0.730 |
| | 3 | 34.21 | **0.691** |
| $\chi^2$ | 1 | 26.7 | 0.745 |
| | 2 | 28.78 | 0.740 |
| | 3 | 29.97 | 0.739 |

Table 3: **Response diversity** of SPPO and RSPO evaluated with Self-BLEU score. Regularization except forward KL improves the diversity.

## 5 RELATED WORK

**Offline RLHF with general divergence for regularization.** The use of general divergence-based regularization has been explored in the context of offline alignment. $f$-DPO (Wang et al., 2023) extends Direct Preference Optimization (Rafailov et al., 2024) from reverse KL regularization to a broader class of $f$-divergences, but primarily demonstrates benefits in generation diversity. The specific effects of individual divergences—and their performance on widely-used benchmarks such as AlpacaEval—remain unexamined. $\chi$PO (Huang et al., 2024) emphasizes the theoretical importance of $\chi^2$ divergence for uncertainty quantification. However, the role of regularization in online iterative preference optimization, particularly its empirical impact on standard benchmarks, has yet to be studied.

**Self-Play Alignment** We emphasize the distinction between our self-play approach and *contrastive* self-play methods including Direct Nash Optimization (DNO) (Rosset et al., 2024) and Iterative Nash Policy Optimization (INPO) (Zhang et al., 2024b). These methods conduct policy optimization with a loss objective necessary but not sufficient for Mirror Descent (MD) update (Beck & Teboulle, 2003). This objective is constructed via winner-loser response comparisons similar to Direct Preference Optimization (DPO) and Identity Preference Optimization (IPO) (Azar et al., 2024). Optimizing such contrastive loss can lead to only an increase in the relative likelihood gap without necessarily enhancing the absolute probability of the preferred response (Pal et al., 2024). In contrast, our method estimates the payoff and directly approximates the MD update by converting it to an equivalent reinforcement learning problem, thereby circumventing the limitations of contrastive approaches.

## 6 CONCLUSION

In this paper, we study the regularization in self-play by proposing a framework, namely Regularized Self-Play Policy Optimization (RSPO). Based on RSPO, we can apply different regularization functions for policy updates by adding the regularization term to the loss functions of existing self-play alignment methods, which we prove is still guaranteed to converge to the NE of the regularized preference optimization game. In the empirical assessments, RSPO with tuned regularizers achieve significant improvement over the base model and unregularized self-play method, SPPO. RSPO significantly improve the performance across various base models, and shows additional parameter-efficiency. We also empirically show that regularization promotes various response quality including diversity. These findings underscore the critical role of regularization as a fundamental component in optimizing self-play alignment.

## 7 ETHICS AND REPRODUCIBILITY STATEMENT

This work raises no question or concern regarding the Code of Ethics. As for reproducibility of our results, we provide details of implementations in Section 4 Implementations and Baselines, and Appendix C.3. We have provided Hyperparameters of each regularization methods in Table 5. All the theoretical results are proved in Appendix A.

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

# A    PROOFS

In this section, we provide detailed assumptions, derivations and proofs of propositions.

**Assumption A.1** (**Relative Convexity of $R$ w.r.t. entropy function**). *We assume the regularization function $R$ of policy $\pi$ is a $1$-strongly convex relative to entropy function. In other words, $\forall \pi, \pi' \in \Delta_{\mathcal{Y}}^{\mathcal{X}}$, and $\psi(\pi) = \langle \pi, \log \pi \rangle$, we have*

$$\langle \partial_\pi R(\pi) - \partial_\pi R(\pi'), \pi - \pi' \rangle \geq \langle \partial_\pi \psi(\pi) - \partial_\pi \psi(\pi'), \pi - \pi' \rangle. \tag{16}$$

Assumption A.1 constrains the class of regularization terms $R$ under which theoretical convergence guarantees can be established. Nonetheless, a broad family of divergences still satisfies this assumption, allowing RSPO to retain convergence properties in a wide range of settings. Among the divergences used in our experiments—including linear combinations—only the forward KL divergence violates this assumption. Interestingly, however, forward KL regularization is empirically observed to reduce response length. To leverage this desirable property while preserving theoretical validity, we propose a linear combination of forward and reverse KL divergences, enabling effective length-controlled generation without sacrificing convergence guarantees, and obtains the best generation quality empirically.

**Assumption A.2** (**On-Policy Assumption**). *We assume that the proposed Regularized Self-Play Policy Optimization (RSPO) method operates in an on-policy manner at each iteration $t$. Formally, we require that sampling $y \sim \pi_\theta$ is well-approximated by sampling $y \sim \pi_t$.*

## A.1    PROOF OF THE EXISTENCE OF REGULARIZED NASH EQUILIBRIUM

**Proposition A.3.** *Nash Equilibrium in the regularized game in Equation* (4) *exists, and it is unique.*

*Proof.* We prove the existence of in this section, largely following the idea of proving the existence of KL regularized Nash Equilibrium by Munos et al. (2023).

Since the utility $u(\pi, \pi')$ is linear in $\pi$ and $\pi'$, and the regularization function is assumed to be convex (Assumption A.1), the regularized preference is concave in $\pi$ and convex in $\pi'$. Therefore, the existence and the uniqueness of a regularized Nash Equilibrium in Equation (4) can be directly derived from the minimax theorem (Sion, 1958).   □

## A.2    PROOF OF RELATIVE CONVEXITY OF COMBINED REGULARIZERS

**Lemma A.4.** *The linear combination of Forward and Reverse KL divergence regularization $R = aKL(\pi \| \mu) + bKL(\mu \| \pi)$, for $a, b > 0$ satisfies that:*

$$\langle \partial_\pi R(\pi) - \partial_\pi R(\pi'), \pi - \pi' \rangle \geq a \langle \partial_\pi \log \pi - \partial_{\pi'} \log \pi', \pi - \pi' \rangle \tag{17}$$

*Proof.* From the definition $R = aKL(\pi \| \mu) + bKL(\mu \| \pi)$, we decompose the left–hand side:

$$\langle \partial_\pi R(\pi) - \partial_\pi R(\pi'), \pi - \pi' \rangle$$
$$= a\langle \partial_\pi KL(\pi \| \mu) - \partial_\pi KL(\pi' \| \mu), \pi - \pi' \rangle + b\langle \partial_\pi KL(\mu \| \pi) - \partial_\pi KL(\mu \| \pi'), \pi - \pi' \rangle$$
$$= a\langle \partial_\pi \log \pi - \partial_{\pi'} \log \pi', \pi - \pi' \rangle + b\langle \partial_\pi KL(\mu \| \pi) - \partial_\pi KL(\mu \| \pi'), \pi - \pi' \rangle$$
$$= a\langle \partial_\pi \psi(\pi) - \partial_\pi \psi(\pi'), \pi - \pi' \rangle + b\langle \partial_\pi KL(\mu \| \pi) - \partial_\pi KL(\mu \| \pi'), \pi - \pi' \rangle.$$

Besides,

$$\langle \partial_\pi KL(\mu \| \pi) - \partial_\pi KL(\mu \| \pi'), \pi - \pi' \rangle = \sum_y (\pi(y) - \pi'(y)) \left( -\frac{\mu(y)}{\pi(y)} + \frac{\mu(y)}{\pi'(y)} \right)$$
$$= \sum_y \frac{\mu(y)\,(\pi'(y) - \pi(y))^2}{\pi(y)\pi'(y)} \geq 0.$$

Therefore,

$$\langle \partial_\pi R(\pi) - \partial_\pi R(\pi'), \pi - \pi' \rangle \geq a \langle \partial_\pi \log \pi - \partial_{\pi'} \log \pi', \pi - \pi' \rangle.$$

□

### A.3 PROOF OF EQUIVALENCE BETWEEN MD AND RSPO

**Proposition A.5.** *Nash-MD and Online Mirror Descent (Munos et al., 2023, Section 6) can be seen as instances of Regularized Self-Play Policy Optimization (RSPO) (Equation* (6)*).*

*Proof.* In this section, we first provide derivations of how Nash-MD is equivalent to RSPO:

$$\nabla_\theta \mathcal{L}_{\text{Nash-MD}} = \nabla_\theta \mathcal{L}_{\text{RSPO}}\Big(\theta; G = \mathbb{P}(y \succ \pi_t^\mu), B = \frac{1}{2}, R = D_{\text{KL}}(\pi_\theta \| \mu)\Big) \tag{18}$$

On one hand, Nash-MD practical loss (Munos et al., 2023, Section 7) is defined as

$$\nabla_\theta \mathcal{L}_{\text{Nash-MD}}(\theta) \tag{19}$$

$$= \mathbb{E}_{\substack{y \sim \pi_\theta, \\ y' \sim \pi_t^\mu}} \Big[ \nabla_\theta \log \pi_\theta(y) \Big( \mathbb{P}(y \succ \pi_t^\mu) - \frac{1}{2} - \tau \log \frac{\pi_\theta(y)}{\mu(y)} \Big) \Big] \tag{20}$$

$$= \mathbb{E}_{\substack{y \sim \pi_\theta, \\ y' \sim \pi_t^\mu}} \Big[ \nabla_\theta \log \pi_\theta(y) \Big( \mathbb{P}(y \succ \pi_t^\mu) - \frac{1}{2} - \tau \log \frac{\pi_\theta(y)}{\mu(y)} + 2\tau \log \frac{\pi_t(y)}{\mu(y)} \Big) \Big] \tag{21}$$

$$= \mathbb{E}_{\substack{y \sim \pi_\theta, \\ y' \sim \pi_t^\mu}} \Big[ \nabla_\theta \log \pi_\theta(y) \Big( \mathbb{P}(y \succ \pi_t^\mu) - \frac{1}{2} - 2\tau \log \frac{\pi_\theta(y)}{\pi_t(y)} + \tau \log \frac{\pi_\theta(y)}{\mu(y)} \Big) \Big] \tag{22}$$

$$= \mathbb{E}_{\substack{y \sim \pi_\theta, \\ y' \sim \pi_t^\mu}} \Big[ \nabla_\theta \log \pi_\theta(y) \Big( \mathbb{P}(y \succ \pi_t^\mu) - \frac{1}{2} - 2\tau \log \frac{\pi_\theta(y)}{\pi_t(y)} \Big) \Big] + \tau \nabla_\theta \mathbb{E}_{\substack{y \sim \pi_\theta, \\ y' \sim \pi_t^\mu}} \Big[ \log \frac{\pi_\theta(y)}{\mu(y)} \Big] \tag{23}$$

$$= \mathbb{E}_{\substack{y \sim \pi_\theta, \\ y' \sim \pi_t^\mu}} \Big[ \nabla_\theta \log \pi_\theta(y) \Big( \mathbb{P}(y \succ \pi_t^\mu) - \frac{1}{2} - 2\tau \log \frac{\pi_\theta(y)}{\pi_t(y)} \Big) \Big] + \tau \nabla_\theta D_{\text{KL}}(\pi_\theta \| \mu) \tag{24}$$

$$= 2\tau^2 \mathbb{E}_{\substack{y \sim \pi_\theta, \\ y' \sim \pi_t^\mu}} \Big[ \nabla_\theta \Big( \log \frac{\pi_\theta(y)}{\pi_t(y)} - \frac{1}{2\tau} \big( \mathbb{P}(y \succ \pi_t^\mu) - \frac{1}{2} \big) \Big)^2 \Big] + \tau \nabla_\theta D_{\text{KL}}(\pi_\theta \| \mu) \tag{25}$$

$$= 2\tau^2 \nabla_\theta \mathbb{E}_{\substack{y \sim \pi_t, \\ y' \sim \pi_t^\mu}} \Big[ \log \frac{\pi_\theta(y)}{\pi_t(y)} - \frac{1}{2\tau} \big( \mathbb{P}(y \succ \pi_t^\mu) - \frac{1}{2} \big) \Big]^2 + \tau \nabla_\theta D_{\text{KL}}(\pi_\theta \| \mu). \tag{26}$$

Equation (20) is the definition of Nash-MD policy gradient. Equation (21) holds because the additional term satisfies that $\mathbb{E}_{\substack{y \sim \pi_\theta, \\ y' \sim \pi_t^\mu}} \big[ \nabla_\theta \log \pi_\theta(y) \big( \log \frac{\pi_t(y)}{\mu(y)} \big) \big] = \nabla_\theta \mathbb{E}_{\substack{y \sim \pi_\theta, \\ y' \sim \pi_t^\mu}} \big[ \log \frac{\pi_t(y)}{\mu(y)} \big] = \nabla_\theta \log \frac{\pi_t(y)}{\mu(y)} = 0$. Equation (24) holds due to the definition of reverse KL divergence. Equation (25) is derived by computing the integral of $\log \pi_\theta(y) \big( \mathbb{P}(y \succ \pi_t^\mu) - \frac{1}{2} - 2\tau \log \frac{\pi_\theta(y)}{\pi_t(y)} \big)$.

On the other hand, we show that Nash-MD and OMD can also be generalized by RSPO *without* external regularization, such that we can add additional regularization to existing regularized self-play methods. Nash-MD practical loss (Munos et al., 2023, Section 7) is defined as

$$\nabla_\theta \mathcal{L}_{\text{Nash-MD}}(\theta) = \mathbb{E}_{\substack{y \sim \pi_\theta, \\ y' \sim \pi_t^\mu}} \Big[ \nabla_\theta \log \pi_\theta(y) \Big( \mathbb{P}(y \succ y') - \frac{1}{2} - \tau \log \frac{\pi_\theta(y)}{\mu(y)} \Big) \Big] \tag{27}$$

$$= \mathbb{E}_{\substack{y \sim \pi_\theta, \\ y' \sim \pi_t^\mu}} \Big[ \nabla_\theta \log \pi_\theta(y) \Big( \mathbb{P}(y \succ y') - \frac{1}{2} - \tau \log \frac{\pi_\theta(y)}{\pi_t(y)} - \tau \log \frac{\pi_t(y)}{\mu(y)} \Big) \Big] \tag{28}$$

$$= \mathbb{E}_{y \sim \pi_\theta} \Big[ \nabla_\theta \log \pi_\theta(y) \Big( \mathbb{P}(y \succ \pi_t^\mu) - \frac{1}{2} - \tau \log \frac{\pi_\theta(y)}{\pi_t(y)} - \tau \log \frac{\pi_t(y)}{\mu(y)} \Big) \Big] \tag{29}$$

$$= \mathbb{E}_{y \sim \pi_t} \Big[ \nabla_\theta \log \pi_\theta(y) \Big( \mathbb{P}(y \succ \pi_t^\mu) - \frac{1}{2} - \tau \log \frac{\pi_\theta(y)}{\pi_t(y)} - \tau \log \frac{\pi_t(y)}{\mu(y)} \Big) \Big] \tag{30}$$

$$= \nabla_\theta \mathbb{E}_{y \sim \pi_t} \Big[ \tau \log \frac{\pi_\theta(y)}{\pi_t(y)} - \Big( \mathbb{P}(y \succ \pi_t^\mu) - \tau \log \frac{\pi_t(y)}{\mu(y)} - \frac{1}{2} \Big) \Big]^2 / 2 \tag{31}$$

$$= \tau^2 \nabla_\theta \mathbb{E}_{y \sim \pi_t} \Big[ \log \frac{\pi_\theta(y)}{\pi_t(y)} - \frac{1}{\tau} \Big( \mathbb{P}(y \succ \pi_t^\mu) - \tau \log \frac{\pi_t(y)}{\mu(y)} - \frac{1}{2} \Big) \Big]^2 / 2. \tag{32}$$

Equation (27) is the definition of practical Nash-MD loss (Munos et al., 2023, Section 7). Equation (28) holds by adding an subtracting the same element $\log \pi_t(y)$. Equation (29) holds due to $\mathbb{E}_{y' \sim \pi_t^\mu} [\mathbb{P}(y \succ y')] = \mathbb{P}(y \succ \pi_t^\mu)$. The learning rate $\eta$ is originally omitted in the paper (Munos et al., 2023). Here Nash-MD is generalized by $\mathcal{L}_{\text{RSPO}}$ with $\eta = \frac{1}{\tau}$ and $R = 0$.

OMD is to execute $\arg\max_\pi \eta\mathbb{E}_{y\sim\pi_t}\left[\mathbb{P}(y\succ\pi_t)-\tau\log\frac{\pi_t(y)}{\mu(y)}\right]-\mathrm{KL}(\pi,\pi_t)$. Therefore, the loss function of the OMD update satisfies

$$\nabla_\theta\mathcal{L}_{\mathrm{OMD}}(\theta)=-\nabla_\theta\eta\mathbb{E}_{y\sim\pi_\theta}\left[\mathbb{P}(y\succ\pi_t)-\tau\log\frac{\pi_t(y)}{\mu(y)}\right]+D_{\mathrm{KL}}(\pi_\theta,\pi_t) \tag{33}$$

$$=-\nabla_\theta\eta\mathbb{E}_{y\sim\pi_\theta}\left[\mathbb{P}(y\succ\pi_t)-\tau\log\frac{\pi_t(y)}{\mu(y)}-\log\frac{\pi_\theta}{\pi_t}\right] \tag{34}$$

$$=\eta\mathbb{E}_{y\sim\pi_\theta}\left[-\nabla_\theta\log\pi_\theta\Big(\mathbb{P}(y\succ\pi_t)-\tau\log\frac{\pi_t(y)}{\mu(y)}-\log\frac{\pi_\theta}{\pi_t}\Big)\right] \tag{35}$$

$$=\frac{\eta}{2}\cdot\mathbb{E}_{y\sim\pi_\theta}\left[\nabla_\theta\Big(\mathbb{P}(y\succ\pi_t)-\tau\log\frac{\pi_t(y)}{\mu(y)}-\log\frac{\pi_\theta(y)}{\pi_t(y)}\Big)^2\right] \tag{36}$$

$$=\frac{\eta}{2}\cdot\mathbb{E}_{y\sim\pi_t}\left[\nabla_\theta\log\frac{\pi_\theta(y)}{\pi_t(y)}-\Big(\mathbb{P}(y\succ\pi_t)-\tau\log\frac{\pi_t(y)}{\mu(y)}\Big)\right]^2. \tag{37}$$

Equation (33) holds because the OMD update is equivalent to descending negative gradient of the feedback $\eta\mathbb{E}_{y\sim\pi}\left[\mathbb{P}(y\succ\pi_t)-\tau\log\frac{\pi_t(y)}{\mu(y)}\right]-\mathrm{KL}(\pi,\pi_t)$. Equation (34) holds due to the definition of $D_{\mathrm{KL}}$. Equation (35) holds by conducting differentiation on multiplication. The remaining equations hold due to simple algebra. Therefore, OMD can also be generalized by RSPO with $G=\mathbb{P}(y\succ\pi_t)-\tau\log\frac{\pi_t(y)}{\mu(y)}$ and without external regularization. $\qquad\square$

### A.4 PROOF OF PROPOSITION 3.1

**Proposition 3.1.** *If $R(\cdot,\mu)$ is 1-strongly convex relative to $\psi$ (Assumption A.1), policy updated by GMMD in Equation (13) has last-iterate convergence to the following Nash Equilibrium of a regularized game:*

$$\max_\pi\min_{\pi'}U(\pi;\pi')-\tau R(\pi,\mu)+\tau R(\pi',\mu). \tag{38}$$

*Proof.* According to Equation (13), GMMD is equivalent to the Algorithm 3.1 in Sokota et al. (2022):

$$z_{t+1}=\arg\min_{z\in\mathcal{Z}}\eta\left(\langle F(z_t),z\rangle+\alpha g(z)\right)+B_\psi(z;z_t), \tag{39}$$

where in our setting, $z=\pi$ is the LLM policy, $F(z_t)=-\partial_\pi U(\pi;\pi_t)$ is the vector of negative partial derivatives of preference w.r.t. each component of $\pi$, $\alpha=\tau$, $g(z)$ is the regularizer $R(\pi)$, and we set $\psi(z)=z\log z$ to convert the Bregman divergence $B_\psi$ to KL divergence. Here $U(\pi;\pi_t)$ is treated as a function of vector form of $\pi$, i.e., $[\pi^0\ \pi^1\ \cdots\ \pi^{|\mathcal{Y}|}]$, thus the gradient is a vector gradient where $\partial_\pi U(\pi;\pi_t)=[\partial U/\partial\pi^0\ \partial U/\partial\pi^1\ \cdots\ \partial U/\partial\pi^{|\mathcal{Y}|}]$.

We then show that in our setting the following assumptions are satisfied. $F$ satisfies that for $\mu>0$ and any $z,z'$, $\langle F(z)-F(z'),z-z'\rangle=0$ since $U$ is linear in $\pi$, and $F(z)-F(z')=-\partial_\pi U(\pi;\pi_t)+\partial_\pi U(\pi';\pi_t)=0$. Therefore, $F$ is Monotone and $L$-smooth. According to Assumption A.1, $g$ is 1-strongly convex relative to $\psi$, i.e., $g(z)\geq g(z')+\frac{g'(z)}{\psi'(z)}(\psi(z)-\psi(z'))$.

Given the assumptions above, according to the Theorem 3.4. in Sokota et al. (2022), the update rule defined in Equation (39) has a last-iterate convergence guarantee to a policy $\pi^*$, which is the solution to the variational inequality problem $\mathrm{VI}(\Delta_\mathcal{Y}^\mathcal{X},F+\alpha\partial g)$, i.e., $\pi^*$ satisfies

$$\langle\partial_\pi\Big(-U(\pi;\pi^*)+\tau R(\pi,\mu)\Big)|_{\pi=\pi^*},\pi-\pi^*\rangle\geq 0,\quad\forall\pi\in\Delta_\mathcal{Y}^\mathcal{X}$$

$$\Leftrightarrow\langle\partial_\pi\Big(-U(\pi;\pi^*)+\tau R(\pi,\mu)-\tau R(\pi^*,\mu)\Big)|_{\pi=\pi^*},\pi-\pi^*\rangle\geq 0,\quad\forall\pi\in\Delta_\mathcal{Y}^\mathcal{X}. \tag{40}$$

Equation (40) indicates that moving from $\pi^*$ towards any direction $\pi-\pi^*$ can not increase the value of the objective preference model $U(\pi;\pi^*)-\tau R(\pi,\mu)+\tau R(\pi^*,\mu)$ at the point of $\pi=\pi^*$, given the opponent is $\pi^*$. Therefore, by symmetry, $\pi^*$ is the Nash Equilibrium of the regularized preference model:

$$\max_\pi\min_{\pi'}U(\pi;\pi')-\tau R(\pi,\mu)+\tau R(\pi',\mu). \tag{41}$$

$\qquad\square$

### A.5 PROOF OF COROLLARY 3.2

*Proof.* We prove that RSPO in Equation (8) is equivalent to GMMD up to multiplying a constant to the gradient, leading to a regularized Nash Equilibrium. We follow SPPO to replace the samples $y \sim \pi_\theta$ with $y \sim \pi_t$ directly since they are equivalent while computing the loss before updating, and rewrite the loss equivalent to GMMD:

$$\nabla_\theta \mathcal{L}_{\text{GMMD}}(\theta) \stackrel{\text{def}}{=} \mathbb{E}_{y \sim \pi_\theta} \left[ \nabla_\theta \log \pi_\theta(y) \left( -\eta G(y; \pi_t) + \log \frac{\pi_\theta(y)}{\pi_t(y)} + B \right) \right] + \tau \nabla_\theta R(\pi_\theta, \mu) \quad (42)$$

$$= \nabla_\theta \left( \frac{1}{2} \mathbb{E}_{y \sim \pi_\theta} \left[ -\eta G(y; \pi_t) + \log \frac{\pi_\theta(y)}{\pi_t(y)} + \eta B \right]^2 + \tau R(\pi_\theta, \mu) \right) \quad (43)$$

$$= \nabla_\theta \left( \frac{1}{2} \mathbb{E}_{y \sim \pi_t} \left[ -\eta G(y; \pi_t) + \log \frac{\pi_\theta(y)}{\pi_t(y)} + \eta B \right]^2 + \tau R(\pi_\theta, \mu) \right) = \frac{1}{2} \nabla_\theta \mathcal{L}_{\text{RSPO}}(\theta) \quad (44)$$

The first equation is the definition of GMMD in Equation (15). The second equation holds due to simple calculus. Equation (44) holds due to the Assumption A.2.

Therefore, according to Equation (44), RSPO is the RL implementation of GMMD, since gradients of losses are equivalent up to multiplying a constant. Then we can derive the convergence guarantee of RSPO by instantiating $G = \mathbb{P}(y \succ \pi_t), B = \frac{1}{2}$ in GMMD

$$\nabla_\theta \mathcal{L}_{\text{RSPO}}(\theta; G = \mathbb{P}(y \succ \pi_t), B = \frac{1}{2})$$

$$= \nabla_\theta \left( \mathbb{E}_{y \sim \pi_t} \left[ \log \frac{\pi_\theta(y)}{\pi_t(y)} - \eta \left( \mathbb{P}(y \succ \pi_t) - \frac{1}{2} \right) \right]^2 + \lambda R(\pi_\theta, \mu) \right) \quad (45)$$

$$= \nabla_\theta \left( \langle \pi_t, , \left( -\eta \partial_\pi \mathbb{P}(\pi \succ \pi_t) + \log \frac{\pi_\theta}{\pi_t} + B \right)^2 \rangle + \lambda R(\pi_\theta, \mu) \right). \quad (46)$$

Equation (45) holds due to definition. Equation (46) holds by treating policy as a vector and rewrite the expectation in vector product form, and $\nabla_\pi \mathbb{P}(\pi \succ \pi_t) \mid_{\pi=\pi_t} = [\mathbb{P}(y^0 \succ \pi_t) \quad \mathbb{P}(y^1 \succ \pi_t) \quad \cdots \quad \mathbb{P}(y^{|\mathcal{Y}|} \succ \pi_t)]^T$, where $y^0, y^1, \cdots, y^{\mathcal{Y}}$ represent all possible values of $y$.

Since in GMMD $G \stackrel{\text{def}}{=} \partial_{\pi(y)} U(\pi; \pi')$ is set to $\mathbb{P}(y \succ \pi_t)$, $U = \mathbb{P}(\pi \succ \pi_t)$ is linear in $\pi$. Thus, according to Proposition 3.1, when the regularizer $R$ satisfies the relative convexity, updating policy following Algorithm 1 with the loss function $\mathcal{L}_{\text{RSPO}}(\theta; G = \mathbb{P}(y \succ \pi_t), B = \frac{1}{2})$ has last-iterate convergence to the Nash Equilibrium of the regularized preference optimization game in Equation (4) by setting $u(\pi; \pi') = \mathbb{P}(\pi \succ \pi')$. □

### A.6 PROOF OF PROPOSITION C.1

*Proof.* $\pi$ is parametrized by $\theta$, $\nabla_\theta D_{\text{KL}}(\pi || \mu) = \mathbb{E}_{\pi_\theta}[\nabla_\theta \log \pi_\theta(y) - \log \mu(y)]^2 / 2$. This is because

$$\nabla_\theta D_{\text{KL}}(\pi || \mu) = \nabla_\theta \sum_y \pi_\theta(y) \cdot (\log \pi_\theta(y) - \log \mu(y)) \quad (47)$$

$$= \sum_y \nabla_\theta \pi_\theta(y) \cdot (\log \pi_\theta(y) - \log \mu(y)) + \sum_y \nabla_\theta \pi_\theta(y)$$

$$= \sum_y \pi_\theta(y) \frac{\nabla_\theta \pi_\theta(y)}{\pi_\theta(y)} \cdot (\log \pi_\theta(y) - \log \mu(y)) + \nabla_\theta \sum_y \pi_\theta(y)$$

$$= \mathbb{E}_{\pi_\theta} [(\log \pi_\theta(y) - \log \mu(y)) \cdot \nabla_\theta (\log \pi_\theta(y) - \log \mu(y))]$$

$$= \mathbb{E}_{\pi_\theta} [\nabla_\theta (\log \pi_\theta(y) - \log \mu(y))^2] / 2. \quad (48)$$

The first equation holds because of the definition of KL divergence. The second equation holds due to applying the product rule of differentiation. The third equation holds due to simple algebra, and the second term will then vanish because of the sum of the probabilities. The fourth equation holds because of simple algebra. □

## A.7 PROOF OF PROPOSITION C.2

*Proof.* $\pi$ is parametrized by $\theta$, then $\nabla_\theta D_{\mathrm{KL}}(\mu||\pi) = \mathbb{E}_\mu[\nabla_\theta \frac{\mu(y)}{\pi_\theta(y)}]$ because

$$\nabla_\theta D_{\mathrm{KL}}(\mu||\pi) = \nabla_\theta \sum_y \mu(y) \cdot (\log \mu(y) - \log \pi_\theta(y)) \tag{49}$$

$$= -\sum_y \mu(y) \nabla_\theta \log \pi_\theta(y) = -\sum_y \pi_\theta(y) \frac{\mu(y)}{\pi_\theta(y)} \nabla_\theta \log \pi_\theta(y)$$

$$= -\mathbb{E}_{\pi_\theta}\left[ \frac{\mu(y) \nabla_\theta \log \pi_\theta(y)}{\pi_\theta(y)} \right] = -\mathbb{E}_{\pi_\theta}\left[ \frac{\mu(y) \nabla_\theta \pi_\theta(y)}{\pi_\theta(y)^2} \right] = \mathbb{E}_{\pi_\theta}\left[ \nabla_\theta \frac{\mu(y)}{\pi_\theta(y)} \right]. \tag{50}$$

The first three equations hold due to the definition of forward KL divergence and simple algebra. The fourth equation comes from rewriting the forward KL following the first three equations. The fifth equation holds by taking the derivative of $\log \pi_\theta$. The sixth equation holds since $\frac{\nabla_\theta \pi_\theta(y)}{\pi_\theta(y)^2} = \nabla_\theta \frac{-1}{\pi_\theta(y)}$. $\qquad\square$

## A.8 PROOF OF PROPOSITION C.3

*Proof.* $\pi$ is parametrized by $\theta$, $\nabla_\theta D_{\chi^2}(\pi_\theta(y)||\mu(y)) = \mathbb{E}_{\pi_\theta}\left[ \frac{\nabla_\theta \pi_\theta(y)}{\mu(y)} \right]$ since

$$D_{\chi^2}(\pi_\theta(y)||\mu(y)) = \frac{1}{2} \sum_y \left( \frac{\pi_\theta(y)}{\mu(y)} - 1 \right)^2 \mu(y) = \frac{1}{2} \sum_y \frac{\pi_\theta(y)^2 - 2\pi_\theta(y)\mu(y) + \mu(y)^2}{\mu(y)}$$

$$= \frac{1}{2} \sum_y \frac{\pi_\theta(y)^2}{\mu(y)} + C(\mu) = \frac{1}{2} \mathbb{E}_{\pi_\theta(y)}\left[ \frac{\pi_\theta(y)}{\mu(y)} \right] + C, \tag{51}$$

where $C(\mu)$ is independent to $\theta$. The first two equations hold according to the definition of Chi-squared divergence. The third equation holds by separating the terms only related to $\mu$ and the term related to $\pi_\theta$. The fourth equation holds by rewriting the summation as the expectation. $\qquad\square$

## B ADDITIONAL RELATED WORK

**Preference Optimization.** Large Language Models (LLMs) recently have obtained remarkable capabilities to accomplish a range of tasks (Jiang et al., 2023a; Dubey et al., 2024; DeepSeek-AI et al., 2025), generating more desirable and helpful content following the user's intention. One of the most important methods to align LLMs with human intentions is Reinforcement Learning from Human Feedback (RLHF), maximizing a preference-based reward penalized by a reverse KL regularization term of the LLM policy and a reference model (Christiano et al., 2017; Ouyang et al., 2022; Rafailov et al., 2024; Azar et al., 2024; Xiong et al., 2024). Since the reference model usually provides safer guidance for policy optimization (Munos et al., 2023), this regularization is crucial in RLHF to prevent over-optimization, which has been extensively studied and extended beyond KL divergence (Wang et al., 2023; Go et al., 2023; Huang et al., 2024). In this work, we instead study the regularization problems in self-play alignment.

**RLHF with General Preference Optimization (Self-Play Alignment).** Azar et al. (2024) introduced the first approach for optimizing LLM policy via general preference models. Nash-MD (Munos et al., 2023) pioneered the application of self-play to general preference optimization by framing it as a two-player game. Subsequent methods have either focused on learning the NE of the original unregularized game (e.g. (Swamy et al., 2024; Wu et al., 2024; Rosset et al., 2024; Wang et al., 2024b)) or the NE of a reverse-KL-regularized preference optimization game (e.g. (Munos et al., 2023; Calandriello et al., 2024; Zhang et al., 2024b)). In contrast, our work explores a broader class of divergence-based regularization techniques for self-play alignment.

Notably, our RSPO can generalize existing self-play methods. Unregularized self-play methods following the preference-based MWU can all be generalized by $\mathcal{L}_{\mathrm{RSPO}}$ without external regularization, and thus can be regularized by simply adding regularization term to the loss functions. Based on the same exponential update rule as in SPPO, SPO (Swamy et al., 2024) is equivalent to updating

policy with the loss in Equation (9). Magnetic Policy Optimization (Wang et al., 2024b), despite incorporating regularization in the policy update, periodically updates $\mu = \pi_t$. Consequently, it inherently follows MWU while incorporating multiple policy updates within each iteration, following

**Online iterative RLHF.** Iterative alignment method incorporates a reliable reward or preference model—including self-play—functions as a self-improving framework by iteratively generating new data using models and optimizing policies based on this data (Schulman et al., 2017; Ouyang et al., 2022; Bai et al., 2022; Touvron et al., 2023; Dong et al., 2024). Moreover, extending powerful offline methods such as DPO to iterative frameworks has led to significant performance gains (Xu et al., 2023; Liu et al., 2023; Tran et al., 2023; Dong et al., 2024; Calandriello et al., 2024; Pang et al., 2024; Xiong et al., 2024; Guo et al., 2024; Tajwar et al., 2024; Cen et al., 2024; Xie et al., 2024). In contrast, our work investigates general preference optimization through self-play from a game-theoretic perspective, shifting the objective from conventional RL optimization to the computation of NE.

## C  ADDITIONAL DETAILS

In this section, we provide additional details of this paper, including the algorithm descriptions of self-play alignment methods, a summarizing table for generalizing existing methods, and our implementation of regularizations.

### C.1  SELF-PLAY ALIGNMENT ALGORITHM

Algorithm 1 shows the overall self-play alignment process. Note that we are sampling $K$ responses per each prompt and obtain pair-wise preferences amongst them for training.

---

**Algorithm 1** Self-Play Alignment

---

**Input:** LLM $\pi_\theta$, preference model $\mathbb{P}$, number of iterations $T$, reference policy $\mu$, loss function for policy update conditioned on utility function $U$: $\mathcal{L}(\theta; U)$, sample size $K$.
**Initialize:** $\pi_0 = \mu$.
**for** $t \in [T]$ **do**
    Sample prompts and responses: $x \sim \mathcal{X}$, $y_{1:K} \sim \pi_t$
    Get pair-wise preferences $u_{ij} = \mathbb{P}(y_i \succ y_j)$, $\forall i, j \in [K]$
    Update policy parameters $\theta = \arg\min_\theta \mathcal{L}(\theta; U)$, $U = [u_{ij}] \in \mathbb{R}^{K \times K}$
    $\pi_{t+1} = \pi_\theta$
**end for**
**Output:** Last-iterate policy $\pi_T$.

---

Specifically, the policy is first initialized as $\pi_0 = \mu$. Then in each iteration $t$, the opponent is set to be the last-iterate policy $\pi_t$ (the reason why it is called self-play), and the responses are sampled from $\pi_t$ (Line 4). The pairwise preferences of the sampled responses are collected using the preference model $\mathbb{P}$ (Line 5). The policy parameters are updated by minimizing a specified loss function $\mathcal{L}(\theta; \mathbb{P})$ based on preferences over responses (Line 6). The loss function $\mathcal{L}(\theta; \mathbb{P})$ is dependent on the inherent online learning method. The main difference between these methods is the choice of loss function $\mathcal{L}(\theta; \mathbb{P})$ applied to the policy update.

### C.2  GENERALIZING EXISTING METHODS

Table 4 shows how the existing methods of self-play alignment can be generalized without external regularization. The algorithms introduced below share the same loss structure as in Equation (6), while their differences present in the update direction $G$, baseline $B$ and the preference model.

### C.3  IMPLEMENTATION OF REGULARIZATION

In practice, accurately estimating the gradient of the regularizer is essential, as many commonly used divergence measures are defined as expectations over $\pi_\theta$. The estimation of divergences has been extensively studied and widely applied in various domains (Rubenstein et al., 2019). For

| Loss | Update Direction ($G$) | Baseline ($B$) | Preference Model |
|---|---|---|---|
| $\mathcal{L}_{\text{SPPO}}$ (Wu et al., 2024) | $\mathbb{P}(y \succ \pi_t)$ | 0.5 | $\mathbb{P}(y \succ y')$ |
| $\mathcal{L}_{\text{OMD}}$ (Munos et al., 2023) | $\mathbb{P}(y \succ \pi_t) - \tau \log \frac{\pi_t(y)}{\mu(y)}$ | Est. | $\mathbb{P}_\tau(y \succ y')$ |
| $\mathcal{L}_{\text{Nash-MD}}$ (Munos et al., 2023) | $\mathbb{P}^\mu(y \succ \pi_t) - \tau \log \frac{\pi_t(y)}{\mu(y)}$ | 0.5 | $\mathbb{P}_\tau(y \succ y')$ |

Table 4: Self-play losses $\mathcal{L}_{\text{RSPO}}$ generalizes different self-play policy optimization methods. $\mathbb{P}^\mu(y \succ \pi_t) = \mathbb{P}(y \succ \pi_t^\mu)$, $\pi_t^\mu$ is the geometric mixture of $\pi_t$ and $\mu$. We abbreviate the estimated baseline that reduce the variance of $G$ the most as est.. $\mathbb{P}_\tau(y \succ y') = \mathbb{P}(y \succ y') - \tau \log \frac{\pi_\theta(y)}{\mu(y)} + \tau \log \frac{\pi'(y')}{\mu(y')}$ is the regularized preference model.

completeness, in this section, we introduce the regularization methods investigated in this study, including Reverse KL, Forward KL, and Chi-Square Divergence.

We begin by deriving the estimation of the Reverse KL divergence based on the following proposition.

**Proposition C.1.** *Reverse KL divergence satisfies:*

$$\nabla_\theta D_{KL}(\pi_\theta || \mu) = \mathbb{E}_{y \sim \pi_\theta}[\nabla_\theta (\log \pi_\theta(y) - \log \mu(y))^2]. \tag{52}$$

According to Proposition C.1, we can estimate the divergence with $\mathbb{E}_{y \sim \pi_\theta}[(\log \pi_\theta(y) - \log \mu(y))^2]$.

We employ two distinct approaches to estimate the forward KL divergence. The first method utilizes importance sampling, referred to as IS-For. KL, and is derived based on the following proposition.

**Proposition C.2.** *The gradient of forward KL divergence satisfies that*

$$\nabla_\theta D_{KL}(\mu || \pi_\theta) = \mathbb{E}_{y \sim \pi_\theta}[\nabla_\theta \mu(y) / \pi_\theta(y)]. \tag{53}$$

Therefore, we can estimate the forward KL divergence by leveraging the expectation $\mathbb{E}_{y \sim \pi_\theta}[\mu(y)/\pi_\theta(y)]$ to estimate the forward KL. Notably, to mitigate the risk of gradient explosion, we apply gradient clipping with a maximum value of 10.

The second method for forward KL is a direct estimation of $D_{\text{KL}}(\mu || \pi_\theta)$. To achieve this, we resample responses from the reference policy $\mu$ using the same prompts from the training dataset, constructing a reference dataset. The KL divergence is then estimated directly based on its definition by uniformly drawing samples from this reference dataset. A key advantage of this approach is that it eliminates the need for importance sampling, as each policy update iteration only requires samples from $\pi_t$.

Similarly, we estimate the Chi-Square divergence using $\mathbb{E}_{y \sim \pi_\theta}[\pi_\theta(y)/\mu(y)]$, based on the following proposition. Due to the presence of the ratio term, Chi-Square divergence estimation also necessitates gradient clipping to prevent instability, for which we set a clip value of 10.

**Proposition C.3.** *Chi-Square divergence has gradient*

$$\nabla_\theta D_{\chi^2}(\pi_\theta || \mu) = \mathbb{E}_{y \sim \pi_\theta}[\nabla_\theta \pi_\theta(y) / \mu(y)]. \tag{54}$$

We also explore the linear combination of different regularization functions to leverage their complementary effects, as in offline RLHF (Huang et al., 2024). The previously established propositions for estimating divergences can still be used in the combined regularization method.

Apart from the flexibility and simplicity of applying different regularization methods, RSPO can generalize existing self-play methods, including the unregularized ones, which enables regularizing off-the-shelf self-play methods in practice with *no change* on their original loss functions or hyperparameters, directly adding an external regularization term to their loss functions.

We then provide the hyperparameters of regularization temperature for each regularizer in our experiments:

# D  ADDITIONAL EXPERIMENTS

In this section, we provide additional experiments, including two synthetic motivating examples and additional results on language tasks.

| Divergence | Parameter(s) |
|---|---|
| Reverse KL (Rev. KL) | $\lambda = 0.5$ |
| Forward K (For. KL) | $\lambda = 1.0$ |
| Chi-Squared ($\chi^2$) | $\lambda = 0.1$ |
| Importance-Sampling Forward KL (IS-For.) | $\lambda = 0.1$ |
| Forward and Reverse KL (IS-For.+Rev. KL) | $\lambda_1 = 0.1, \lambda_2 = 0.5$ |

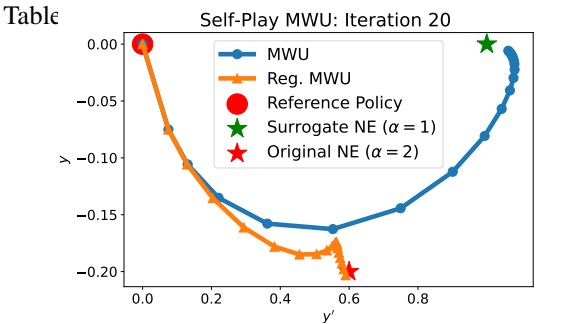

Figure 5: Motivating Example: 20 iterations of MWU and regularized MWU with the same learning rate to solve saddle point problem $\max_y \min_{y'} f(y, y', \alpha)$, where $f(y, y'; \alpha) = \frac{\alpha}{2}y'^2 + (y' - 1)(y - 1) - \frac{\alpha}{2}y^2$, first introduced in (Sokota et al., 2022). We assume that we only have access to a misspecified (surrogate) preference $f(y, y'; \alpha = 1)$, while the ground truth human preference is $f(y, y'; \alpha = 2)$.

## D.1 Regularization in Game Solving

The regularization in the preference model is not used in all game-theoretic self-play methods. Here we investigate the necessity of regularization and offer a motivating example in Figure 5, a saddle point solving problem $\min_x \max_y \frac{\alpha}{2}x^2 + (x - 1)(y - 1) - \frac{\alpha}{2}y^2$. There exists a reference point as the initial values of $x$ and $y$. We assume that both reference point and the Nash Equilibrium (NE) of the surrogate preference model (Surrogate NE) are close to the original NE but on different sides of the original NE.

Typically, the surrogate preference/reward models are not positively related to the reference policy. Thus, it is a reasonable abstracted example of NLHF by treating the reference point as reference policy and surrogate NE as the optimal policy obtained by optimizing the surrogate preference/reward. The results of the 20 iterations self-play MWU with an early stopping show that regularization can be used to prevent reward over-optimization (reaching surrogate NE). A well-tuned regularization leads to faster convergence to the unknown original NE. Thus, regularization can be effective in preventing over-optimization in self-play.

## D.2 Diversity on 2D Example

We offer an analysis of our method compared to unregularized self-play (SPPO) on a 2D example in Figure 6. The area with a darker color is assigned a higher reward value. We use the preference defined by the $L^2$ norm between two actions. We also set the reference policy to be uniform. According to the figure, the unregularized method tends to converge to a single point on the manifold of the large reward. While regularized methods have diverse sampled actions.

## D.3 More Results on AlpacaEval-2.0 and PairRM

In Figure 7 and Table 6, we present further results of RSPO evaluated using AlpacaEval. As presented in Figure 7, mixed regularization of the forward and reverse KL resulted in the best performance, while its average response length did not exceed that of reverse KL-only regularization. When compared to various other well-known baselines including GPT-4 and Claude, RSPO-trained model initialized from Mistral-7B shows notable performance, outperforming GPT-4 0314 and LLaMA

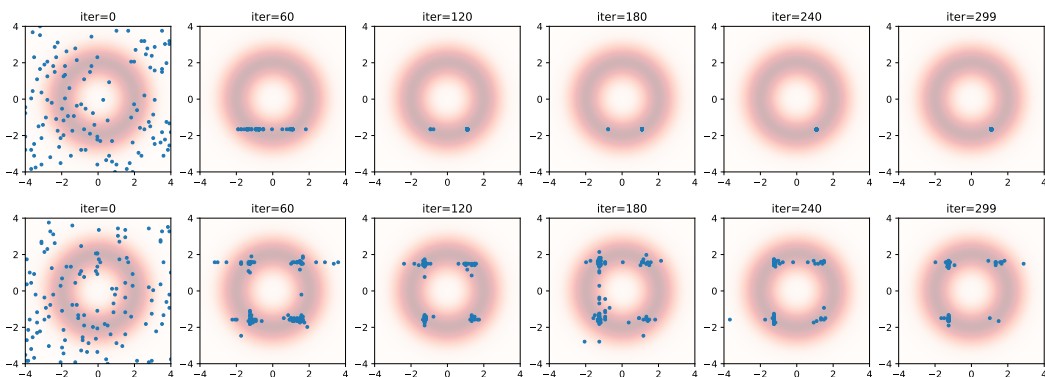

Figure 6: Samples in a 2D example of different iterations of SPPO (top) and RSPO (bottom) with external forward KL regularization to a uniform random reference policy. SPPO added simple external regularization that can generate multi-modal policies.

3 70B Instruct in LCWR. When response lengths are ignored, our RSPO-trained 7B model even outperforms Claude 3 Opus.

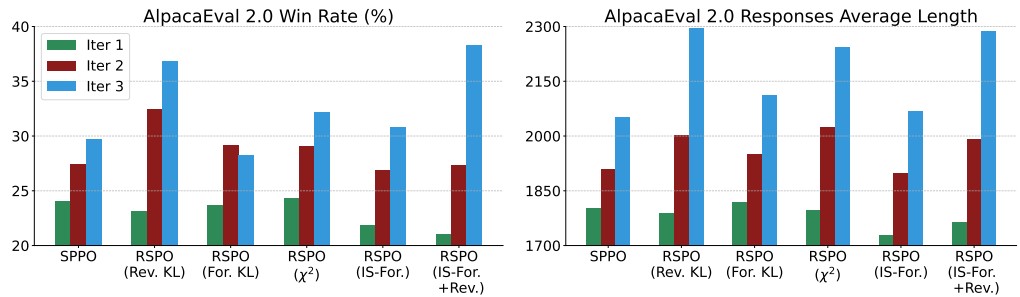

Figure 7: Win rates and the average length of SPPO and RSPO with different regularization methods. From left to right, regularization methods: Reverse KL, Forward KL, Chi-Squared, Importance-Sampling Forward KL, Importance-Sampling Forward, and Reverse KL linear combination.

Table 6: **Left:** AlpacaEval-2.0 performance of **RSPO** with different regularization temperatures. **Right:** AlpacaEval-2.0 performance comparison with popular models. Our model, Mistral-7B-RSPO Iter3, outperforms GPT-4 0314 and LLaMA 3 70B Instruct in LCWR. When only win rate is considered, our model even outperforms Claude 3 Opus.

| Regularization Temperature | Iter | LCWR (%) | WR (%) | Length |
|---|---|---|---|---|
| forward: 0.1 reverse: 0.5 | 1 | 23.16 | 21.06 | 1763 |
| forward: 0.1 reverse: 0.5 | 2 | 27.91 | 27.38 | 1992 |
| forward: 0.1 reverse: 0.5 | 3 | **35.44** | 38.31 | 2286 |
| forward: 0.01 reverse: 0.5 | 1 | 24.63 | 22.57 | 1793 |
| forward: 0.01 reverse: 0.5 | 2 | 28.21 | 28.56 | 2006 |
| forward: 0.01 reverse: 0.5 | 3 | 32.24 | 36.77 | 2411 |

| Model | AlpacaEval 2.0 | |
|---|---|---|
| | LC. Win Rate | Win Rate |
| GPT-4 Turbo | 50.0 | 50.0 |
| Claude 3 Opus | 40.5 | 29.1 |
| Mistral-7B-RSPO Iter3 | 35.44 | 38.31 |
| GPT-4 0314 | 35.3 | 22.1 |
| LLaMA 3 70B Instruct | 34.4 | 33.2 |
| GPT-4 0613 | 30.2 | 15.8 |
| Mistral Medium | 28.6 | 21.9 |
| Mistral-7B-SPPO Iter3 | 28.5 | 31.0 |

# E  OTHERS

In this section, we provide other details including additional related works, compute resources, societal impacts and limitations.

## E.1  COMPUTE RESOURCES

We conduct experiments on $8 \times$A100 80GB for training and single A100 80GB for evaluation.

Table 7: **Left: SPPO replication** Iteration-wise LCWR, WR, and Length results. Overoptimization exists according to the results. **Right:** Pairwise win rate of RSPO on Ultrafeedback validation set rated by pairRM. RSPO has higher win rates against all the baselines.

| Iter | LCWR (%) | WR (%) | Length |
|------|----------|--------|--------|
| 1 | 26.36 | 24.04 | 1802 |
| 2 | 28.38 | 27.43 | 1909 |
| 3 | 29.17 | 29.75 | 2051 |
| 4 | 28.45 | 30.20 | 2257 |
| 5 | 27.93 | 30.11 | 2301 |
| 6 | 28.03 | 30.99 | 2435 |
| 7 | 25.46 | 28.25 | 2471 |
| 8 | 22.94 | 28.26 | 2691 |
| 9 | 24.47 | 28.57 | 3402 |

| Methods | RSPO (IS-For.+Rev.) Iter3 Win Rate |
|---------|-------------------------------------|
| **Snorkel (Iterative-DPO)** | 0.55 |
| **SPPO Iter3** | 0.57 |
| **SimPO** | 0.50 |

Table 8: **RSPO on LLaMA3-8B-Instruct:** Comparison of SPPO and RSPO iterations on AlpacaEval LCWR and Avg. Len.

| Base: LLaMA3-8B-Instruct | AlpacaEval LCWR | Avg. Len |
|--------------------------|-----------------|----------|
| SPPO-Iter1 | 31.73 | 1962 |
| SPPO-Iter2 | 35.15 | 2021 |
| SPPO-Iter3 | 38.77 | 2066 |
| RSPO (Rev. KL, 0.5)-Iter1 | 31.04 | 2100 |
| RSPO (Rev. KL, 0.5)-Iter2 | 35.89 | 2289 |
| **RSPO (Rev. KL, 0.5)-Iter3** | **43.66** | **2504** |

Table 9: **RSPO on Gemma-2-2B-IT:** Comparison of SPPO and RSPO iterations on AlpacaEval LCWR and Avg. Len for Gemma-2-2B-IT.

| Base: Gemma-2-2B-IT | AlpacaEval LCWR | Avg. Len |
|---------------------|-----------------|----------|
| SPPO-Iter1 | 43.68 | 2191 |
| SPPO-Iter2 | 49.24 | 2372 |
| SPPO-Iter3 | 50.54 | 2471 |
| RSPO (Rev. KL, 0.1)-Iter1 | 41.87 | 2213 |
| RSPO (Rev. KL, 0.1)-Iter2 | 48.94 | 2337 |
| RSPO (Rev. KL, 0.1)-Iter3 | 51.83 | 2443 |

Table 10: **Regularization Effect of RSPO on LLaMA3-8B-Instruct:** Forward KL regularization effect on AlpacaEval LCWR and Avg. Len. Increasing the strength of For. KL regularization only brings slight reduction in average length.

| Method | AlpacaEval LCWR | Avg. Len |
|--------|-----------------|----------|
| SPPO-Iter1 | 31.73 | 1962 |
| SPPO-Iter2 | 35.15 | 2021 |
| SPPO-Iter3 | 38.77 | 2066 |
| RSPO (IS-For. KL, **0.01**, clip 10)-Iter1 | 30.87 | 1939 |
| RSPO (IS-For. KL, **0.01**, clip 10)-Iter2 | 33.58 | 1962 |
| RSPO (IS-For. KL, **0.01**, clip 10)-Iter3 | 36.80 | 1998 |
| RSPO (IS-For. KL, **0.1**, clip 10)-Iter1 | 26.72 | 1920 |
| RSPO (IS-For. KL, **0.1**, clip 10)-Iter2 | 32.22 | 1973 |
| RSPO (IS-For. KL, **0.1**, clip 10)-Iter3 | 36.64 | 1948 |

### E.2 SOCIETAL IMPACTS

This study introduces a novel framework for fine-tuning large language models through self-play, incorporating regularization toward a reference model. Ethical considerations may emerge if the reference model exhibits harmful behaviors, or if the preference model used for policy updates

Table 11: **Regularization Effect of RSPO on LLaMA3-8B-Instruct:** Due to only slight length reduction in length as shown in Table 10, the combination regularization of For. and Rev. KL shows less improvement compared to only regularized with Rev. KL. However, the performance is still superior compared to SPPO.

| Method | AlpacaEval LCWR | Avg. Len |
|---|---|---|
| SPPO-Iter1 | 31.73 | 1962 |
| SPPO-Iter2 | 35.15 | 2021 |
| SPPO-Iter3 | 38.77 | 2066 |
| RSPO (Rev. KL, 0.5)-Iter1 | 31.04 | 2100 |
| RSPO (Rev. KL, 0.5)-Iter2 | 35.89 | 2289 |
| **RSPO (Rev. KL, 0.5)-Iter3** | **43.66** | **2504** |
| RSPO (IS-For. + Rev. KL)-Iter1 | 30.89 | 2100 |
| RSPO (IS-For. + Rev. KL)-Iter2 | 34.94 | 2308 |
| RSPO (IS-For. + Rev. KL)-Iter3 | 41.84 | 2465 |

Table 12: **ArmoRM Evaluation.** Evaluation of diverse response quality aspects of fine-tuned Mistral-7B model on Ultrafeedback validation set using ArmoRM Wang et al. (2024a). The combined application of forward and reverse KL regularization leads to superior performance compared to either form of regularization applied independently.

| Methods | Overall Score | Instruction Following | Truthfulness | Honesty | Helpfulness |
|---|---|---|---|---|---|
| Snorkel | 0.706 | 0.781 | 0.796 | 0.821 | 0.760 |
| SPPO | 0.716 | 0.798 | 0.812 | **0.836** | 0.771 |
| RSPO ($\chi^2, \lambda = 0.1$) | 0.713 | 0.793 | 0.805 | 0.827 | 0.769 |
| RSPO (Rev. $\lambda = 0.5$) | 0.718 | 0.798 | 0.805 | 0.831 | **0.773** |
| RSPO (Rev. $\lambda = 1$) | 0.715 | 0.798 | 0.807 | 0.826 | 0.769 |
| RSPO (For. $\lambda = 0.1$) | 0.711 | 0.795 | 0.809 | 0.824 | 0.760 |
| RSPO (For. $\lambda = 0.5$) | 0.713 | 0.793 | 0.815 | 0.826 | 0.749 |
| **RSPO (For.+Rev.)** | **0.719** | **0.805** | **0.816** | 0.833 | 0.768 |

inadvertently assigns higher ratings to harmful outputs. However, drawing on prior research, we find no evidence that the proposed approach poses direct negative societal impacts.

### E.3 LIMITATIONS

A theoretical limitation lies in the nature of the regularization term $R$ which is required to be relatively convex with respect to entropy (Assumption A.1). Both reverse KL divergence and $\chi^2$ divergence satisfy this property, whereas forward KL divergence does not. This discrepancy is evident in performance metrics such as raw win rates. Interestingly, forward KL has a beneficial side effect of reducing response length. To leverage the length reduction and reconcile the decreasing win rate, we adopt a linear combination of forward and reverse KL divergences—an approach that not only satisfies the relative convexity condition but also exploits the complementary strengths of each to achieve improved control over response length while maintaining theoretical soundness.

### E.4 RELATED WORK

Incorporating regularization into equilibrium-finding procedures has been extensively studied and shown to improve convergence guarantees across a range of algorithmic game-theoretic settings (Facchinei & Pang, 2003; Hennes et al., 2019; Cen et al., 2021; Perolat et al., 2021; Abe et al., 2022; Pattathil et al., 2023; Abe et al., 2023; 2024), including methods tailored to extensive-form games (Liu et al., 2022). However, these approaches have primarily been evaluated in small-scale problem settings, leaving limited evidence that they can be directly applied to or effectively scaled for large games such as those arising in large language model (LLM) preference optimization. Prior work that does scale to large games (Perolat et al., 2022; Jacob et al., 2022) remains restricted to reverse-KL regularization. In contrast, our work scales reinforcement-learning–based preference optimization to large games while accommodating general forms of regularization. Moreover, closely

related methods such as (Abe et al., 2023; 2024) implicitly constrain the regularizer to be linear in the policy, whereas our approach—grounded in the theory by Sokota et al. (2022)—supports general relatively-convex regularizers.

