# OpenReview forum: "RSPO: Regularized Self-Play Alignment of Large Language Models"
_ICLR.cc/2026/Conference — Submitted to ICLR 2026_

### Official Review · Reviewer_8aye · 2025-10-21

**Soundness:** 3
**Presentation:** 3
**Contribution:** 1
**Rating:** 2
**Confidence:** 4

**Summary:**

This paper introduces Regularized Self-Play Policy Optimization (RSPO), a new framework for aligning large language models (LLMs) using self-play. The authors argue that existing self-play alignment methods have not adequately explored regularization against a reference policy, which is critical for preventing over-optimization. RSPO addresses this by providing a flexible and simple way to incorporate various "plug-and-play" regularizers into the self-play loss function. The framework is theoretically grounded, with proofs showing that it maintains convergence to a Nash Equilibrium of the regularized game. Empirically, the paper demonstrates that RSPO with well-chosen regularizers significantly improves performance over unregularized baselines (like SPPO) across several benchmarks (AlpacaEval-2, Arena-Hard, MT-Bench) and base models (Mistral-7B, LLaMA-8B, Gemma-2B). The authors also analyze the distinct effects of different regularizers (e.g., forward vs. reverse KL) and show that their combination can lead to the best results.

**Strengths:**

- The paper does an excellent job of motivating the need for better regularization in self-play alignment. The problem of over-optimization, especially with imperfect preference models, is well-known in RLHF, and its insufficient investigation in the context of self-play is a relevant research gap.

- The proposed RSPO framework is elegant in its simplicity. The idea of adding a modular, plug-and-play regularization term to an existing self-play loss is appealing. Crucially, this simplicity does not come at the cost of theoretical rigor. The authors provide solid theoretical guarantees, linking RSPO to Mirror Descent and proving last-iterate convergence, which adds significant credibility to the method.

- The experimental setup is thorough and convincing. The authors evaluate RSPO on multiple popular benchmarks and against strong baselines. The use of different base models demonstrates the general applicability of the method. The ablation studies on different regularizers and their temperatures (Figure 4) provide valuable insights into their distinct effects on win rate and response length.

- The results presented are impressive. RSPO consistently outperforms the unregularized SPPO baseline and other methods across the board. The significant win rate improvements on benchmarks like AlpacaEval-2 (e.g., 28.5% to 35.4% for Mistral-7B) clearly demonstrate the practical benefits of the proposed approach.

**Weaknesses:**

In my opinion, this paper has a major and unavoidable issue. **All the theories in this paper (e.g., RSPO is flexible for general regularization) have already been thoroughly studied in the field of game theory**, yet the authors introduce them into the LLM field without any citations. Admittedly, the presentation of this paper is excellent. However, directly copying results from the game theory field contributes nothing to the community.

[1] Kenshi Abe, Mitsuki Sakamoto, Kaito Ariu, Atsushi Iwasaki. Boosting Perturbed Gradient Ascent for Last-Iterate Convergence in Games ICLR 2025.

[2] Kenshi Abe, Kaito Ariu, Mitsuki Sakamoto, Atsushi Iwasaki. Adaptively Perturbed Mirror Descent for Learning in Games. ICML 2024.

**Questions:**

Please see weaknesses.

---

> ### Author Response · Authors · 2025-11-20
> **Response to Reviewer 8aye**
>
> The major issue proposed by the reviewer is a fundamental misunderstanding regarding the scope and positioning of our work. This paper is centered on **alignment problems in large language models**, as clearly indicated in the title, and employs self-play-based methods as a technical approach.
>
> Our primary contributions to the LLM alignment community include **theoretical unification of existing game-theoretic alignment methods**, and providing a **simple, flexible, scalable, and theoretically sound framework** to add **general regularization** for self-play alignment. Our aim is **not to develop a new game theory algorithm**, but to develop a novel scalable and robust self-play (game-theoretic) approach for LLMs preference optimization. In this work, we specifically focus on the critical and important term in RLHF or preference optimization—the regularization w.r.t. the reference model. We provide detailed responses to the subpoints of the concern.
>
> * We **strongly disagree** with the claim that "authors introduce theory into the LLM field without any citations"—**this is factually incorrect**. We have appropriately cited foundational work: Freund & Schapire (1997) for Multiplicative Weights Update (lines 119-120) and the Magnetic Mirror Descent algorithm (lines 243-245, line 838, line 851), which is a critical work to our paper. The references listed by the reviewer have no relation to this topic.
> * We **strongly disagree** with the reviewer's claim on "copying results from the game theory field." **Our first critical theoretical result (Section 3.1) is not about game theory at all**—it is the **unification of existing mainstream self-play-based LLM alignment methods**, which makes the integration of general regularization simple, flexible, and scalable. Our convergence guarantee (Section 3.2) is indeed based on the theory of Magnetic Mirror Descent. However, we **properly cite it and adapt it to the setting of preference optimization**. Adapting theoretical tools to new domains with proper attribution is standard research practice and constitutes legitimate contribution, not "copying."
> * We **strongly disagree** with the assertion that our work "contributes nothing to the community." Our proposed framework unifies existing mainstream self-play methods, paving the way to incorporate general regularization w.r.t. the reference model. More importantly, our extensive empirical investigations on LLMs reveal **novel findings** regarding the effects of different regularization schemes across various base models and preference models. These practical insights are directly valuable to researchers and practitioners working on LLM alignment. To claim "zero contribution" dismisses our theoretical analysis and substantial experimental work and empirical discoveries that advance understanding in this active research area.
>
> **We respectfully but firmly request the reviewer reconsider this assessment in light of our clarifications.**
>
> ---
>
> Additionally, the reviewer states that we have directly copied results from the game theory literature and cites references \[1\] and \[2\]. Could the reviewer please clarify which specific results from these works \[1-2\] you believe we have copied? If certain results indeed transfer to our setting, we are happy to acknowledge and appropriately cite them. From our current understanding, the references provided are not directly related.
>
> \[1\] Kenshi Abe, Mitsuki Sakamoto, Kaito Ariu, Atsushi Iwasaki. Boosting Perturbed Gradient Ascent for Last-Iterate Convergence in Games ICLR 2025\.
>
> \[2\] Kenshi Abe, Kaito Ariu, Mitsuki Sakamoto, Atsushi Iwasaki. Adaptively Perturbed Mirror Descent for Learning in Games. ICML 2024\.

---

> > ### Comment · Reviewer_8aye · 2025-11-20
> >
> > I apologize for the oversight; I omitted a crucial reference, which is [3].
> >
> > Regarding your proposal to "add general regularization for self-play alignment," this concept has already been established in [3]. Specifically, it is outlined in their Section 3.1. The introduction of regularization to the utility function, as illustrated in your Figure 1, is entirely equivalent to their framework. Furthermore, Section 5 in [3] addresses the aspect of convergence.
> >
> > As to the claim that highly relevant literature has been cited, this is unfortunately incorrect. MWU algorithm is fundamental. Regarding Magnetic Mirror Descent, I suggest a literature review. There exists a substantial body of contemporary work employing regularization to achieve last-iterate convergence across various game settings, including normal-form games (you consider), extensive-form games, and mean-field games. [1,2,3] have already conducted thorough investigations on this topic.
> >
> > In my personal assessment, the algorithm most pertinent to your work is actually the ICML version of Deep Nash [4]. Although its theoretical foundation does not hold in discrete time settings, [3] has successfully addressed this issue.
> >
> > [3] Abe, Kenshi, Kaito Ariu, Mitsuki Sakamoto, and Atsushi Iwasaki. "Adaptively Perturbed Mirror Descent for Learning in Games."
> >
> > [4] Perolat, Julien, Remi Munos, Jean-Baptiste Lespiau, Shayegan Omidshafiei, Mark Rowland, Pedro Ortega, Neil Burch et al. "From Poincaré Recurrence to Convergence in Imperfect Information Games: Finding Equilibrium via Regularization."

---

> > > ### Author Response · Authors · 2025-11-24
> > > **Response to Reviewer 8aye**
> > >
> > > Thanks for the swift response.
> > >
> > > We realize the reviewer has a very critical misunderstanding—the reviewer thinks our motivation for adding regularization is to ensure last-iterate convergence. However, as clearly explained in the abstract, introduction, and the main manuscript, **regularization is a practical choice in RLHF to address over-optimization**. However, current self-play alignment approaches lack a mechanism to mitigate the reward over-optimization problem. It is an interesting coincidence that this regularization term has both theoretical effect (convergence guarantee) and empirical meaning (addressing over-optimization in alignment).
> > >
> > > **We would also like to point out that none of the references provided by the reviewer focus on real LLM alignment; rather, they address general game-solving and evaluate in small-scale games.** The reviewer's previous judgment that “contributing nothing to the community” points to the game theory community. However, the scope and position of this paper are LLM alignment.
> > >
> > > **Thus, we believe the reviewer judges our work from an incorrect perspective. Our contributions should be evaluated by comparison with other LLM self-play alignment works (e.g., \[7–11\]) rather than the whole algorithmic game theory world.** Otherwise, the judgment becomes unfair, and the same criticism can be used to judge \[7-11\] as well. The unfairness of the reviewer’s judgment is also reflected by omitting our empirical contributions completely.
> > >
> > > &nbsp;
> > >
> > > > Q1: The concept "add general regularization for self-play alignment" has already been established in \[3\]. Specifically, it is outlined in their Section 3.1. The introduction of regularization to the utility function, as illustrated in your Figure 1, is entirely equivalent to their framework. Furthermore, Section 5 in \[3\] addresses the aspect of convergence.
> > >
> > > **A1:**
> > >
> > > First of all, \[3\] refers to the same literature as \[2\] in your previous comments. Would you mind clarifying why you thought \[3\] was omitted in your review? If this was a mistake, please let us know the reference you intended to use. For now, we can only assume the reviewer is referring to \[2\] again. We provide the following clarifications:
> > >
> > > * Reference \[3\] is not about alignment at all, but is purely an algorithmic game theory paper. The terms “alignment,” “LLM,” and “preference” never appear in the paper. Thus, it is obviously incorrect to claim that \[3\] has established the concept of regularized self-play alignment.
> > > * We also notice the algorithm in \[3\] is fundamentally different from ours. In \[3\] Section 3.1 Eq. 3,  the regularization term ($\langle G,x\rangle$) is required to be linear in $x$ as $G$ is independent of $x$, thus being more restrictive compared to ours. Our method is based on Magnetic Mirror Descent (MMD) (see \[5\], Algorithm 3.1), the regularization g(z) is not limited to linear ones. Therefore [3] is entirely different.
> > >
> > > * MMD (ICLR 2023\) is prior to \[3\] (ICML 2024). MMD introduced general regularization in Eq. (2) along with its convergence guarantee, and discussed how it can be scaled using deep reinforcement learning. MMD also provided discrete-time convergence. Therefore, our citation of MMD as the highly relevant work is “correct”.
> > >
> > > * The existence of \[3\] or even MMD does not diminish the novelty or contribution of our work. Implementing self-play with LLMs is far from trivial compared with small-scale games such as in \[3\]. Our method cleverly avoids implementing a full self-play algorithm from scratch to add general regularization, feasible to leverage off-the-shelf methods and implementations directly. The convergence guarantee is included solely to ensure theoretical rigor.
> > >
> > > &nbsp;
> > >
> > > > Q2: As to the claim that highly relevant literature (e.g. MWU and MMD) has been cited, this is unfortunately incorrect.
> > >
> > > **A2:** We cited MWU when introducing SPPO because the self-play LLM alignment method SPPO was developed based on this fundamental algorithm, as written in their paper. Our implementation is largely based on SPPO (as we have stated in the paper), so it makes no sense to claim SPPO (MWU) is not highly relevant.
> > >
> > > As explained in A1, MMD is the most relevant work to our paper. The extended discussion and connection to deep reinforcement learning established in the MMD paper is directly useful for our work, compared to \[3\].

---

> > > > ### Author Response · Authors · 2025-11-24
> > > > **Response to Reviewer 8aye**
> > > >
> > > > > Q3: Regarding Magnetic Mirror Descent, I suggest a literature review. There exists a substantial body of contemporary work employing regularization to achieve last-iterate convergence across various game settings, including normal-form games (you consider), extensive-form games, and mean-field games. \[1,2,3\] have already conducted thorough investigations on this topic.
> > > >
> > > > **A3:** We agree that there is a large body of work on regularization. But as we clarified at the beginning, the motivation and the scope is not algorithmic game theory. Still, we augment our literature reviews on the works mentioned by the reviewer in Appendix E.4.
> > > >
> > > > &nbsp;
> > > >
> > > >
> > > > > Q4: In my personal assessment, the algorithm most pertinent to your work is actually the ICML version of Deep Nash \[4\]. Although its theoretical foundation does not hold in discrete time settings, \[3\] has successfully addressed this issue.
> > > >
> > > > **A4**: The assessment is unfortunately incorrect. Firstly, DeepNash is not \[4\]. More importantly, both DeepNash \[6\] and \[4\] are restricted to reverse KL divergence for regularization. As explained in A1, the most pertinent work is still MMD.
> > > >
> > > > &nbsp;
> > > >
> > > > ---
> > > > **Reference**
> > > >
> > > > \[3\] Abe, Kenshi, Kaito Ariu, Mitsuki Sakamoto, and Atsushi Iwasaki. "Adaptively Perturbed Mirror Descent for Learning in Games."
> > > >
> > > > \[4\] Perolat, Julien, Remi Munos, Jean-Baptiste Lespiau, Shayegan Omidshafiei, Mark Rowland, Pedro Ortega, Neil Burch et al. "From Poincaré Recurrence to Convergence in Imperfect Information Games: Finding Equilibrium via Regularization."
> > > >
> > > > \[5\] Sokota, Samuel, et al. "A unified approach to reinforcement learning, quantal response equilibria, and two-player zero-sum games." arXiv preprint arXiv:2206.05825 (2022).
> > > >
> > > > \[6\] Perolat, Julien, et al. "Mastering the game of Stratego with model-free multiagent reinforcement learning." Science 378.6623 (2022): 990-996.
> > > >
> > > > \[7\] Munos, Rémi, et al. "Nash learning from human feedback." Forty-first International Conference on Machine Learning. 2024\.
> > > >
> > > > \[8\] Wu, Yue, et al. "Self-play preference optimization for language model alignment." arXiv preprint arXiv:2405.00675 (2024).
> > > >
> > > > \[9\] Wang, Mingzhi, et al. "Magnetic preference optimization: Achieving last-iterate convergence for language model alignment." arXiv preprint arXiv:2410.16714 (2024).
> > > >
> > > > \[10\] Zhang, Yuheng, et al. "Improving LLM general preference alignment via optimistic online mirror descent." arXiv preprint arXiv:2502.16852 (2025).
> > > >
> > > > \[11\] Zhang, Yuheng, et al. "Iterative nash policy optimization: Aligning llms with general preferences via no-regret learning." arXiv preprint arXiv:2407.00617 (2024).

---

### Official Review · Reviewer_ge5C · 2025-10-30

**Soundness:** 3
**Presentation:** 4
**Contribution:** 4
**Rating:** 8
**Confidence:** 3

**Summary:**

This paper proposes a framework called Regularized Self-Playing Strategy Optimization (RSPO) for alignment fine-tuning of Large Language Models (LLMs). Existing self-play alignment methods have not sufficiently explored regularization terms to mitigate “over-optimization” issues. RSPO unifies existing self-play approaches by introducing a plug-and-play regularization term while ensuring the algorithm converges to the Nash equilibrium of the corresponding regularized game. Experimental results demonstrate that RSPO with appropriately chosen regularization terms significantly outperforms the unregularized baseline model (SPPO) and achieves performance improvements across multiple models.

**Strengths:**

1. Theoretically rigorous and experimentally comprehensive, it identifies and resolves the critical issue of “over-optimization.”
2. As a practical framework for LLM alignment, RSPO demonstrates excellent flexibility and scalability, laying the groundwork for subsequent exploration of more complex regularization methods.

**Weaknesses:**

1. Diversity evaluation (self-BLEU) is limited to a single metric and lacks complementary measures such as entropy.
2. While the baseline comparison includes mainstream methods like SimPO, it does not evaluate other recent variants. We recommend expanding the comparison to enhance persuasiveness.

**Questions:**

1. In the experiment, only PairRM (0.4B) was used as the preference model. Has RSPO performance been tested with larger preference models (e.g., 70B RM)? If the preference model is more accurate, does the need for regularization decrease?
2. Could you provide λ tuning details, such as the search range, to facilitate reproduction?
3. Computational overhead (Appendix E.3) remains unquantified. Could you provide training times for RSPO vs. SPPO for reference?

---

> ### Author Response · Authors · 2025-11-20
> **Response to Reviewer ge5C**
>
> We sincerely appreciate the valuable evaluation and recognition of our work’s theoretical contributions and extensive experiments, as well as our method’s flexibility and scalability. We address the concerns as follows.
>
> &nbsp;
>
> > Q1: Diversity evaluation (self-BLEU) is limited to a single metric and lacks complementary measures such as entropy.
>
> **A1**: We agree that entropy can serve as a meaningful metric when appropriately normalized by generation length. However, because self-BLEU is specifically designed to assess linguistic diversity, we adopt it as our representative metric.
>
> &nbsp;
>
> > Q2: While the baseline comparison includes mainstream methods like SimPO, it does not evaluate other recent variants. We recommend expanding the comparison to enhance persuasiveness.
>
> **A2**: Thanks for pointing this out. We include three new baselines EXPO \[2\], AlphaDO \[3\], and POET \[4\] in the following table.
>
> | Model | AlpacaEval LCWR |
> | :---- | :---- |
> | **RSPO-Gemma-2B-PairRM** | **51.8%** |
> | AlphaDPO-LLaMA-8B | 46.6% |
> | **RSPO-LLaMA-8B-PairRM** | **43.7%** |
> | **RSPO-Mistral7B-PairRM** | **35.4%** |
> | POET \+ SimPO-LLaMA-8B | 33.8% |
> | AlphaDPO-Mistral7B | 32.3% |
> | ExPO \+ SPPO-Mistral7B-PairRM | 31.8% |
> | POET \+ SimPO-Mistral7B | 25.3% |
>
> &nbsp;
>
> > Q3: In the experiment, only PairRM (0.4B) was used as the preference model. Has RSPO performance been tested with larger preference models (e.g., 70B RM)? If the preference model is more accurate, does the need for regularization decrease?
>
> **A3**: We have tested using a larger preference model, GPM-8B \[1\] (see Figure 3). While the performance improvement is reduced compared to smaller preference models, RSPO still demonstrates **superior parameter efficiency**: fine-tuning LLaMA-8B with our method using pairRM-0.5B achieves comparable performance to fine-tuning LLaMA-8B with the significantly larger GPM-8B with SPPO.
>
> &nbsp;
>
> > Q4: Could you provide λ tuning details, such as the search range, to facilitate reproduction?
>
> **A4**: We conduct a hyperparameter search over λ∈{0.01,0.05,0.1,0.5,1,2}. For the importance-sampling forward KL baseline, we stabilize training by clipping importance sampling weights at thresholds of 5 and 10. Regarding the metrics used for hyperparameter tuning, both pairRM and ArmoRM are effective.
>
> &nbsp;
>
> > Q5: Computational overhead (Appendix E.3) remains unquantified. Could you provide training times for RSPO vs. SPPO for reference?
>
> **A5**:
>
> **Response sampling costs** are identical between the two methods, as both require the same number of samples for loss computation.
>
> **Training costs** are also equivalent. Computing the regularization term requires only the log probabilities of the completions, which are already computed for the primary loss. Therefore, no additional forward passes are needed, resulting in **no additional training time overhead**.
>
> ---
>
> \[1\] Yifan Zhang, Ge Zhang, Yue Wu, Kangping Xu, and Quanquan Gu. Beyond bradley-terry models: A general preference model for language model alignment. arXiv preprint arXiv:2410.02197, 2024a.
>
> \[2\] Zheng, Chujie, et al. "Model extrapolation expedites alignment." Proceedings of the 63rd Annual Meeting of the Association for Computational Linguistics (Volume 1: Long Papers). 2025\.
>
> \[3\] Wu, Junkang, et al. "AlphaDPO: Adaptive Reward Margin for Direct Preference Optimization." Forty-second International Conference on Machine Learning.
>
> \[4\] Xiao, Zeguan, et al. "Towards Bridging the Reward-Generation Gap in Direct Alignment Algorithms." arXiv preprint arXiv:2506.09457 (2025).

---

### Official Review · Reviewer_ZdsE · 2025-10-30

**Soundness:** 2
**Presentation:** 3
**Contribution:** 2
**Rating:** 2
**Confidence:** 4

**Summary:**

The authors propose Regularized Self-Play Policy Optimization (RSPO), a novel framework that flexibly allows for "plug-and-play" integration of general regularization terms into the standard self-play loss. The paper claims to provide theoretical guarantees for this approach, asserting that RSPO is an implementation of Generalized Magnetic Mirror Descent (GMMD) and thus achieves last-iterate convergence to the Nash Equilibrium of the corresponding regularized game. Empirically, RSPO is shown to significantly outperform the unregularized SPPO baseline across various models and benchmarks.

**Strengths:**

1. The paper tackles the significant and practical issue of over-optimization in state-of-the-art self-play alignment methods (like SPPO), which largely lack the explicit regularization needed for stable, long-run training.

2. A key strength of RSPO is its simplicity. It allows researchers to add any regularization term ($\lambda R(\pi_{\theta}, \mu)$) directly to the existing SPPO loss, making it highly practical and easy to implement for experimenting with diverse regularizers (e.g., forward KL, reverse KL, $\chi^2$).

3. Extensive experiments showing that RSPO significantly outperforms the unregularized SPPO baseline on multiple benchmarks (AlpacaEval-2, Arena-Hard). It empirically proves its ability to solve the over-optimization problem (Figure 2, left) and provides valuable in-depth analysis on how different regularizers distinctly impact model behavior.

**Weaknesses:**

1. The claimed theoretical equivalence between RSPO and GMMD (Section 3.2, Appendix A.4) is invalid. In Appendix A.4 (Eq.41--46), the authors attempt to prove that $\nabla L_{RSPO} \propto \nabla L_{GMMD}$, but Eq.41 already defines $\nabla L_{GMMD} = \frac{1}{2} \nabla L_{RSPO}$; thus, the proof is circular and does not demonstrate any real equivalence between the two algorithms. In addition, the substitution $y \sim \pi_\theta \to y \sim \pi_t$ is made without importance-sampling correction or a small-step-size assumption, introducing unacknowledged bias. Since RSPO modifies the SPPO MSE surrogate by adding a regularizer, while GMMD performs a single mirror-descent proximal update, they are different optimization procedures. Consequently, the last-iterate convergence guarantee of GMMD cannot be directly inherited by RSPO, which should instead be described as a heuristic method inspired by GMMD rather than a theoretically equivalent one.
2. Even if we accept the GMMD proof, Prop. 3.1 hinges on Assumption A.1 (1-relative strong convexity w.r.t. entropy). The paper explicitly notes that Forward KL violates this assumption, yet the best-performing Mistral-7B setup uses IS-For.+Rev. KL. No sufficient conditions are provided to ensure that the mixed regularizer remains relatively strongly convex. Hence, the theoretical guarantee does not apply to the paper’s strongest empirical results.

**Questions:**

See weaknesses

---

> ### Author Response · Authors · 2025-11-20
> **Response to Reviewer ZdsE**
>
> We thank the reviewer for the insightful and detailed evaluation. We have addressed all concerns below.
>
> &nbsp;
>
> > Q1: The claimed theoretical equivalence between RSPO and GMMD (Section 3.2, Appendix A.5) is invalid. In addition, the substitution is made without importance-sampling correction or a small-step-size assumption, introducing unacknowledged bias.
>
> **A1:** We believe there is a misunderstanding regarding the proof structure. For clarity, we have revised the proof in Appendix A.5, adding Equations 42-44 to establish the equivalence between RSPO and GMMD step-by-step under the below stated assumptions.
>
> We have additionally introduced a small-step-size assumption in Assumption A.2. This assumption is standard in policy gradient analyses. Incorporating importance sampling would be suboptimal in practice, as importance weights can produce excessively large gradient norms when the denominator is close to zero. This happens in our importance-sampling forward KL (fKL) regularization. Mitigating such instability typically requires additional heuristic techniques—such as aggressive gradient clipping—which further complicate the optimization procedure.
>
> ---
>
> &nbsp;
>
> > Q2: RSPO modifies the SPPO by adding a regularizer, while GMMD performs a single mirror-descent proximal update, they are different optimization procedures. Consequently, the last-iterate convergence guarantee of GMMD cannot be directly inherited by RSPO.
>
> **A2:** RSPO is a general framework that generalizes both SPPO and Nash-MD with arbitrary regularization; it is not merely SPPO augmented with a regularizer. From the perspective of the optimization procedure, the distinction between GMMD and SPPO with regularization is relatively minor—principally, GMMD uses a single policy update per iteration, whereas SPPO employs multiple updates. As discussed in our earlier response A1, under a single policy update the small-step-size assumption holds, i.e., $y \\sim \\pi\_t \\leftrightarrow y \\sim \\pi\_\\theta$​, which implies that the gradients of the GMMD and RSPO (implemented by SPPO with regularization) losses are equivalent. More importantly, as shown in Eq. (10) in Sec. 3.1, RSPO can also be instantiated directly from Nash-MD by adding or replacing regularizers, yielding an objective that is exactly equivalent to GMMD. **Therefore, we maintain that our theoretical contribution—establishing convergence guarantees for RSPO with general regularizers under Assumption A.2—is both valid and significant.**
>
> In our experiments, we indeed adopt the multi-update implementation to better utilize samples, since sampling is computationally expensive for LLM-based RL. This choice is a practical compromise specific to the LLM fine-tuning setting. This mirrors the relationship between vanilla Policy Gradient and Proximal Policy Optimization (PPO), where additional update steps improve sample efficiency in practice. Nevertheless, we believe that this implementation choice—and the resulting gap between theory and practice—should not diminish the significance of the theoretical contribution or the value of this work.

---

> ### Author Response · Authors · 2025-11-20
> **Response to Reviewer ZdsE**
>
> > Q3: Prop. 3.1 hinges on Assumption A.1 (1-relative strong convexity w.r.t. entropy). The paper explicitly notes that Forward KL violates this assumption, yet the best-performing Mistral-7B setup uses IS-For.+Rev. KL. No sufficient conditions are provided to ensure that the mixed regularizer remains relatively strongly convex.
>
> **A3**: We would like to provide two points:
>
> - The strong performance of combined forward and reverse KL regularization is only observed on Mistral-7B fine-tuned models. However, across all other models (Figure 3), reverse KL consistently performs as the superior regularization, demonstrating robustness.
> - **We provide new analysis on the sufficient conditions for the mixed regularizers in Appendix A.2 Lemma A.4.** Notably, applying forward KL alone yields degraded performance—only the linear combination is effective. More importantly, while forward KL itself violates the relatively convex assumption (A.1), the linear combination of forward and reverse KL satisfies the inequality (Eq. 16) in Assumption A.1 up to a multiplicative constant. With this property, the last-iterate convergence still holds by setting appropriate learning temperature $\eta$ (please refer to Lemma D.4. and Appendix D.2 proof of Theorem 3..4 in [1]).
>
> ---
>
> \[1\] Sokota, S., D'Orazio, R., Kolter, J. Z., Loizou, N., Lanctot, M., Mitliagkas, I., ... & Kroer, C. (2022). A unified approach to reinforcement learning, quantal response equilibria, and two-player zero-sum games. arXiv preprint arXiv:2206.05825.

---

### Meta-Review · Area_Chair_7iCE · 2026-01-02

**Summary:**

The main concerns driving my suggested decision are theoretical credibility, especially the gap between the stated guarantees and the practical training setup, and novelty relative to existing game-theoretic results and methods. While the work is overall strong and attractive, the "plug-and-play regularization" framing appears primarily practically useful and may constitute more of an engineering contribution. Reviewers raised substantial concerns that the claimed equivalence to GMMD and the inherited last-iterate convergence guarantees are not adequately justified under the actual multi-update implementation and the best-performing regularizers. Also, the novelty critique argues that key ideas are largely established in prior game-theory literature and may be under-cited, making the contribution feel more like adaptation/translation than a fundamental advance. Overall, there remains clear disagreement between reviewers and authors regarding both novelty and the convincingness of the theoretical claims. If the authors can address these misunderstandings thoroughly, especially by tightening the theoretical claims to match practice and strengthening the prior-art discussion, the revised paper would likely be very strong.

**Reviewer Concerns:**

Concerns addressed by the rebuttal (at least partially):

1. The authors added a small-step-size condition and revised the proof structure, which helps clarify the intended assumptions.

2. They expanded comparisons (e.g., EXPO/AlphaDPO/POET), provided the $\lambda$ search range, tested a larger preference model, and argued minimal overhead, addressing most of reviewer ge5C’s reproducibility and baseline concerns.

3. They clarified that the paper’s scope is LLM alignment, not proposing new game-theory algorithms, which helps contextualize reviewer 8aye’s concern.

Concerns are still outstanding:

1. The extent of the theory–practice gap, especially whether the convergence guarantees meaningfully apply to the practical multi-update setting (ZdsE).

2. The assumption mismatch for forward/mixed KL and whether the strongest empirical configuration is actually covered by the stated sufficient conditions (ZdsE).

3. The sufficiency of novelty and citation coverage relative to recent regularization-for-convergence/game-solving work; the rebuttal frames the contribution as domain adaptation, but the "what is new" boundary remains unclear (8aye).

**Reviewer Scores:**

Reviewer ZdsE: unchanged, or +1 if convinced the revised proof and added assumptions adequately narrow the claim; otherwise remains 2 due to the theory–practice mismatch.

Reviewer ge5C: unchanged, or +1 after the added baselines, tuning details, and larger model test.

Reviewer 8aye: unchanged; the follow-up comments continued to raise concerns about prior art and citation coverage, even after the rebuttal.

---

### Decision · Program_Chairs · 2026-01-26

Reject